# *Lasiodiplodia mitidjana* sp. nov. and other Botryosphaeriaceae species causing branch canker and dieback of *Citrus sinensis* in Algeria

Akila Berraf-Tebbal[1]*, Alla Eddine Mahamedi[2,3], Wassila Aigoun-Mouhous[2,4], Milan Špetík[1], Jana Čechová[1], Robert Pokluda[1], Miroslav Baránek[1], Aleš Eichmeier[1], Artur Alves[5]

1 Mendeleum—Institute of Genetics, Faculty of Horticulture, Mendel University in Brno, Lednice, Czech Republic, 2 Laboratoire de Biologie des Systèmes Microbiens (LBSM), Département des Sciences Naturelles, Ecole Normale Supérieure de Kouba-Alger, Alger, Algeria, 3 Département de Biologie, Faculté des Sciences de la Nature et de la Vie et Sciences de la Terre, Université de Ghardaïa, Ghardaïa, Algeria, 4 Département des Biotechnologies, Faculté des Sciences de la Nature et de la Vie, Université de Blida, Blida, Algeria, 5 Departamento de Biologia, CESAM, Universidade de Aveiro, Aveiro, Portugal

* qqberraf@mendelu.cz

**Data Availability Statement:** All relevant data are within the manuscript and its Supporting Information files.

## Abstract

Several Botryosphaeriaceae species are known to occur worldwide, causing dieback, canker and fruit rot on various hosts. Surveys conducted in ten commercial citrus orchards in the northern region of Algeria revealed five species of Botryosphaeriaceae belonging to three genera associated with diseased trees. Morphological and cultural characteristics as well as phylogenetic analyses of the internal transcribed spacer (ITS) region and the translation elongation factor 1-alpha (*tef1*-α) identified *Diplodia mutila*, *Diplodia seriata*, *Dothiorella viticola*, *Lasiodiplodia mediterranea* and a novel species which is here described as *Lasiodiplodia mithidjana* sp. nov.. Of these, *L. mithidjana* (14.1% of the samples) and *L. mediterranea* (13% of the samples) were the most widespread and abundant species. Pathogenicity tests revealed that *L. mediterranea* and *D. seriata* were the most aggressive species on citrus shoots. This study highlights the importance of Botryosphaeriaceae species as agents of canker and dieback of citrus trees in Algeria.

## Introduction

Citrus represent one of the main fruit crops in the world. They are widely recognized by their nutritional and health-related properties of both fresh fruit and juice. Being produced in more than 140 countries, citrus production reached more than 13 million Tonnes, in 2018 [1]. Citrus cultivation is one of the major contributors to the Algerian wealth and is part of the traditional agriculture of the country. Many types of citrus are grown in Algeria, including oranges (48 400 ha), clementines (10 817 ha), mandarins (2 347 ha), lemons (4 409 ha) and grapefruits (83 ha) [2]. Despite the high adaptation capacity of citrus trees to different climates [3], a number of unfavourable factors has led to a decrease of the total citrus yield in Algeria. Among these factors, ageing trees, droughts, inappropriate cultural practices and the effects of various

**Funding:** This work was supported by Technologická Agentura České Republiky (TJ02000096) and Ministerstvo Školství, Mládeže a Tělovýchovy (CZ.02.1.01./0.0/0.0/16_017/ 0002334). A. Alves acknowledges the financial support from FCT/MCTES to CESAM (UID/AMB/ 50017/2019), through national funds.

**Competing interests:** The authors have declared that no competing interests exist.

pests and pathogens are the most important [3, 4]. Citrus diseases are numerous and diverse, and are caused by phytopathogenic agents belonging to viruses, viroids, phytoplasmas, bacteria, and fungi [3] Some pathogens cause very serious diseases, predisposing to, and inciting dieback, while others are less serious [3–6].

Recently, trunk diseases have become a growing threat in both, old and newly established orchards of citrus, worldwide. Symptoms include leaves that become yellow and fall early, shoots and twigs die, increasing the risk of citrus decay as the damage expands to the trunk [3, 7–9]. To date, among the fungi that impact citrus, *Diaporthe* species are well known for causing stem-end rot and melanose of fruits, young leaf and shoot gummosis and blight of perennial branches and trunks, in Greece, Italy, Malta, Portugal, Spain, China, Korea, New Zealand, and the USA [9–13].

*Fusarium* and *Neocosmospora* have also been reported causing canker and dieback diseases of citrus, in Tunisia, Greece, Italy and Spain [14–16]. The Diatrypaceae are other canker and dieback pathogens impacting citrus orchards [17]. Several *Eutypella* spp. have been reported from *Citrus* sp. In southern California desert, three distinct species of *Eutypella* are found associated with citrus branch canker, namely: *Eutypella citricola*, *E. microtheca* and a *Eutypella* sp. [17–19].

In addition to the above fungal pathogens that compromise citrus crops, several Botryosphaeriaceae species are known to colonize citrus trees. The Botryosphaeriaceae family is recognized as an important and widely distributed plant pathogen, which impacts on a variety of economically important hosts. It comprises at least 24 genera encompassing 222 known species, living as endophytes, saprobes, or plant pathogens [20, 21]. Recent studies carried out in California, Italy and Tunisia have highlighted the Botryosphaeriaceae as the most prevalent fungi that cause cankers, vascular necrosis and dieback of citrus trees [8, 17, 18, 22]. Adesemoye et al. [23] recovered various *Botryosphariaceae* species from necrotic tissues of citrus branch canker and rootstock, including, *Diplodia seriata*, *D. mutila*, *Dothiorella viticola*, *Doth. iberica*, *Lasiodiplodia parva*, *Neofusicoccum australe*, *N. luteum*, *N. mediterraneum*, *N. parvum* and *Neoscytalidium dimidiatum*. In Iran, Abdollahzadeh et al. [24], described *Lasiodiplodia citricola* from citrus trees showing symptoms of branch dieback.

In Algeria, members of the Botryosphaeriaceae have been reported to cause diseases on *Vitis vinifera* [25–27], *Quercus suber* [28] and *Cupressus macrocarpa* [29]. Linaldeddu et al. [30] isolated and described *L. mediterranea* from a cankered branch of *Citrus sinensis* trees in northern Algeria. However, the impact of Botryosphaeriaceae species on citrus trees in Algeria has not been studied in detail. Therefore, the aim of this study was to investigate and determine the incidence as well as evaluating the pathogenicity of the Botryosphaeriaceae species associated with branch canker and dieback in the major citrus-growing region of Algeria.

## Materials and methods

### Ethics statement

No specific permits were required for the described field studies. This study did not involve endangered or protected species.

### Field survey and sampling

Surveys were conducted in ten commercial orchards in the northern region of Algeria. Specifically, in the Mitidja plain at the base of the Tell Atlas Mountains. The sampling was done in four municipalities; namely Oued El Aleug (4 orchards), Chiffa (2 orchards), Boufarik (2 orchards) and the coastal town, Sidi Fredj, located within the territory of the Staoueli municipality, situated by the Mediterranean Sea (2 orchards). The field diagnosis and sampling were

**Table 1. Citrus orchards surveyed and number of samples collected.**

| Locality | Orchards | Area (ha) | Number of trees sampled | Number of samples processed |
|---|---|---|---|---|
| Oued El Alleug | a | 18 | 5 | 9 |
| | b | 16 | 5 | 6 |
| | c | 28 | 5 | 7 |
| | d | 6.8 | 5 | 6 |
| Chiffa | a | 25 | 10 | 13 |
| | b | 18 | 10 | 10 |
| Boufarik | a | 43 | 10 | 10 |
| | b | 27 | 10 | 11 |
| Staoueli | a | 32 | 10 | 10 |
| | b | 15 | 10 | 10 |

performed between April 2013 and March 2015. Samples were collected from the orchards with permission of landowners. All the prospected orchards had approximately the same age (between 25 and 30 years old). Trunks and branches showing symptoms such as dead shoots, defoliation, cankers, wood necrosis, and dieback were collected, randomly. A total of 80 symptomatic sweet orange (*Citrus sinensis*) trees were sampled (Table 1).

## Fungal isolation and morphological characterization

In the laboratory, all samples were processed by peeling the outer bark surface with a sterilized scalpel. Longitudinal and transversal cuts were made to reveal the type and localization of the internal necrosis. From each lesion detected, ten pieces of wood, approx. 5 mm$^2$, were cut from the margins between necrotic and healthy tissues. These pieces were submerged in 4% sodium hypochlorite for 15 min, washed thrice with sterile distilled water, dried with sterilized filter paper and placed onto the surface of potato dextrose agar (PDA, Difco Laboratories). Plates were incubated at 25°C until growth was detected. The mycelium emerging from wood pieces were transferred onto fresh PDA plates and incubated under the same conditions.

Preliminary identifications to genus and tentative species level were based on colony and conidial morphology (colony colour, colony growth pattern, conidial size, shape, colour, striation, septation, conidiogenous cells, and presence of paraphyses) according to Phillips et al. [20]. Isolates that lacked pycnidia production on PDA were placed on autoclaved pine needles in ¼ strength PDA within 2–3 weeks, incubated at 25°C under mixed near-UV and cool-white fluorescent light in a 12 h light 12 h dark regime for 2–6 weeks, to enhance fruiting body production. Conidiogenous layer and conidia were mounted in 100% lactic acid and observed with a Nikon 80i light microscope.

## DNA extraction, PCR amplification and sequencing

Total genomic DNA was extracted from 7 days old axenic cultures, grown on PDA at 25°C, following Santos and Phillips [31]. PCR reactions were carried out with *Taq* DNA polymerase, nucleotides and buffers supplied by MBI Fermentas (Vilnius, Lithuania). PCR reaction mixtures were prepared as previously described by Alves et al. [32], with the addition of 5% DMSO to improve the amplification of some difficult DNA templates. The ITS region plus D1/D2 domain of the LSU was amplified with the primer pair ITS1 [33] and NL4 [34]. The amplification conditions were initial denaturation of 5 min at 95°C, followed by 29 cycles of 30 s at 94°C, 30 s at 50°C, and 1.5 min at 72°C, and a final extension of 10 min at 72°C. Part of the translation elongation factor 1 alpha gene (*tef1-α*) was amplified with primers EF1-688F and

EF1-1251R [35]. The amplification conditions were: initial denaturation of 5 min at 95 C, followed by 30 cycles of 30 s at 94˚C, 45 s at 55˚C, 1½ min at 72˚C, and a final extension period of 10 min at 72˚C. ITS and *tef1*-α regions were sequenced in both directions by STAB Vida Lda (Portugal), using the Sanger method.

The nucleotide sequences were read and edited with BioEdit Alignment Editor V.7.0.9.0 [36]. Newly generated sequences were deposited in GenBank (Table 2). Homological sequences of the newly sequenced ones were retrieved from the GenBank using the Basic Local Alignment Search Tool (BLAST) [37].

## Phylogenetic analysis

Sequences of all *Lasiodiplodia* species known from culture were retrieved from GenBank (S1 Table) and aligned with sequences of the isolates obtained in this study. Alignments were done with ClustalX v. 1.83 [38] using the following parameters: pairwise alignment parameters (gap opening = 10, gap extension = 0.1) and multiple alignment parameters (gap opening = 10, gap extension = 0.2, transition weight = 0.5, delay divergent sequences = 25%). Alignments were checked and manual adjustments made if necessary using BioEdit v. 7.2.5 [36]. Maximum Likelihood (ML) and Maximum Parsimony (MP) analyses were performed using MEGAX [39]. The best fitting DNA evolution model was determined also by MEGAX. A discrete Gamma distribution with five categories was used to model evolutionary rate differences among sites (+G, parameter = 0,3394). ML analysis was performed on a Neighbour-Joining starting tree automatically generated by the software. Nearest-Neighbour-Interchange (NNI) was used as the heuristic method for tree inference. MP analysis was done using the Tree-Bisection-Regrafting (TBR) algorithm with search level 1 in which the initial trees were obtained by the random addition of sequences (10 replicates). The robustness of the trees (ML and MP) was evaluated by 1000 bootstrap replications.

## Pathogenicity test

Aggressiveness of the fungi was evaluated by measuring the lengths of the internal lesions. The ability of the isolates to cause cankers was assessed *in vivo* on 1-year-old detached shoots collected from symptomless *Citrus sinensis* trees, following Hamrouni et al. [22] and Adesemoye et al. [39]. From each phylogenetically resolved species, two representative isolates were selected. The shoots with 25 mm in diameter were cut into equal length (25 cm long). They were then surface disinfected with 70% ethanol and wounded on an intermediate internode, with a scalpel. From each strain, a 5 mm diameter mycelial plug taken from a 5- day old colony growing on PDA was placed into the wound. Negative controls were inoculated with fresh, non-colonized, PDA plugs. The point of inoculation was covered with wet sterile cotton and sealed with Parafilm® to prevent desiccation. Subsequently, the cuttings were well watered and maintained under favorable conditions. There were 10 replicates per isolate, and the same number of cuttings was used as controls. One month after inoculation, lengths of lesions produced by each strain were measured. In an attempt to recover the inoculated fungi and complete Koch's postulates, necrotic tissue from the margin of the lesions was taken and placed onto PDA.

## Statistical analyses

Internal lesion lengths from the pathogenicity test were analysed. Means where checked for normality using Shapiro-Wilk test with α = 0.05, then differences in lesion lengths caused by fungal isolates belonging to different species were assessed using One-way analysis of variance (ANOVA) with P ≤ 0.05. Significant differences with the confidence interval of 95% were

**Table 2. Botryosphaeriaceae species included in this study.**

| Species | Isolate number | Host/ Substrate | Origin | Collector | GenBank accession numbers | |
|---|---|---|---|---|---|---|
| | | | | | ITS | *tef1-α* |
| *Lasiodiplodia mediterranea* | ALG104 | *Citrus*/wood canker | Algeria, Boufarik | Akila Berraf-Tebbal | MN104094 | MN159093 |
| *L. mediterranea* | ALG105 | *Citrus*/wood canker | Algeria, Oued El Alleug | Akila Berraf-Tebbal | MN104095 | MN159094 |
| *L. mediterranea* | ALG40 | *Citrus*/wood canker | Algeria, Oued El Alleug | Akila Berraf-Tebbal | MN104096 | MN159095 |
| *L. mediterranea* | ALG78 | *Citrus*/wood canker | Algeria, Oued El Alleug | Akila Berraf-Tebbal | MN104097 | MN159096 |
| *L. mediterranea* | ALG41 | *Citrus*/wood canker | Algeria, Oued El Alleug | Akila Berraf-Tebbal | MN104098 | MN159097 |
| *L. mediterranea* | ALG36 | *Citrus*/wood canker | Algeria, Oued El Alleug | Akila Berraf-Tebbal | MN104099 | MN159098 |
| *L. mediterranea* | ALG80 | *Citrus*/wood canker | Algeria, Oued El Alleug | Akila Berraf-Tebbal | MN104100 | MN159099 |
| *L. mediterranea* | ALG106 | *Citrus*/wood canker | Algeria, Boufarik | Akila Berraf-Tebbal | MN104101 | MN159100 |
| *L. mediterranea* | ALG107 | *Citrus*/wood canker | Algeria, Boufarik | Akila Berraf-Tebbal | MN104102 | MN159101 |
| *L. mediterranea* | ALG108 | *Citrus*/wood canker | Algeria, Boufarik | Akila Berraf-Tebbal | MN104103 | MN159102 |
| *L. mediterranea* | CBS 124060 | *Vitis*, wood fragment | Italy, Sicily | S. Burruano | KX464148 | MN938928 |
| **L. mitidjana** | ALG81 | *Citrus*/wood canker | Algeria, Oued El Alleug | Akila Berraf-Tebbal | MN104104 | MN159103 |
| **L. mitidjana** | ALG44 | *Citrus*/wood canker | Algeria, Oued El Alleug | Akila Berraf-Tebbal | MN104105 | MN159104 |
| **L. mitidjana** | ALG39 | *Citrus*/wood canker | Algeria, Oued El Alleug | Akila Berraf-Tebbal | MN104106 | MN159105 |
| **L. mitidjana** | ALG42 | *Citrus*/wood canker | Algeria, Oued El Alleug | Akila Berraf-Tebbal | MN104107 | MN159106 |
| **L. mitidjana** | ALG38 | *Citrus*/wood canker | Algeria, Oued El Alleug | Akila Berraf-Tebbal | MN104108 | MN159107 |
| **L. mitidjana** | ALG43 | *Citrus*/wood canker | Algeria, Oued El Alleug | Akila Berraf-Tebbal | MN104109 | MN159108 |
| **L. mitidjana** | ALG37 | *Citrus*/wood canker | Algeria, Oued El Alleug | Akila Berraf-Tebbal | MN104110 | MN159109 |
| **L. mitidjana** | ALG34 | *Citrus*/wood canker | Algeria, Oued El Alleug | Akila Berraf-Tebbal | MN104111 | MN159110 |
| **L. mitidjana** | ALG82 | *Citrus*/wood canker | Algeria, Oued El Alleug | Akila Berraf-Tebbal | MN104112 | MN159111 |
| **L. mitidjana** | ALG109 | *Citrus*/wood canker | Algeria, Boufarik | Akila Berraf-Tebbal | MN104113 | MN159112 |
| **L. mitidjana** | ALG110 | *Citrus*/wood canker | Algeria, Boufarik | Akila Berraf-Tebbal | MN104114 | MN159113 |
| **L. mitidjana** | ALG111 = MUM 19.90 | *Citrus*/wood canker | Algeria, Boufarik | Akila Berraf-Tebbal | MN104115 | MN159114 |
| **L. mitidjana** | ALG112 | *Citrus*/wood canker | Algeria, Boufarik | Akila Berraf-Tebbal | MN104116 | MN159115 |
| *Diplodia seriata* | ALG93 | *Citrus*/wood canker | Algeria, Staoueli | Akila Berraf-Tebbal | MN104117 | MN159116 |
| *D. seriata* | ALG94 | *Citrus*/wood canker | Algeria, Staoueli | Akila Berraf-Tebbal | MN104118 | MN159117 |
| *D. seriata* | ALG98 | *Citrus*/wood canker | Algeria, Staoueli | Akila Berraf-Tebbal | MN104119 | MN159118 |
| *D. seriata* | ALG91 | *Citrus*/wood canker | Algeria, Chiffa | Akila Berraf-Tebbal | MN104120 | MN159119 |
| *D. seriata* | ALG92 | *Citrus*/wood canker | Algeria, Staoueli | Akila Berraf-Tebbal | MN104121 | MN159120 |
| *D. seriata* | ALG90 | *Citrus*/wood canker | Algeria, Chiffa | Akila Berraf-Tebbal | MN104122 | MN159121 |
| *D. seriata* | ALG89 | *Citrus*/wood canker | Algeria, Chiffa | Akila Berraf-Tebbal | MN104123 | MN159122 |
| *D. seriata* | ALG96 | *Citrus*/wood canker | Algeria, Staoueli | Akila Berraf-Tebbal | MN104124 | MN159123 |
| *D. seriata* | ALG95 | *Citrus*/wood canker | Algeria, Staoueli | Akila Berraf-Tebbal | MN104125 | MN159124 |
| *D. seriata* | ALG97 | *Citrus*/wood canker | Algeria, Staoueli | Akila Berraf-Tebbal | MN104126 | MN159125 |
| *D. mutila* | ALG99 | *Citrus*/wood canker | Algeria, Chiffa | Akila Berraf-Tebbal | MN104127 | MN159126 |
| *D. mutila* | ALG103 | *Citrus*/wood canker | Algeria, Chiffa | Akila Berraf-Tebbal | MN104128 | MN159127 |
| *D. mutila* | ALG100 | *Citrus*/wood canker | Algeria, Chiffa | Akila Berraf-Tebbal | MN104129 | MN159128 |
| *D. mutila* | ALG102 | *Citrus*/wood canker | Algeria, Chiffa | Akila Berraf-Tebbal | MN104130 | MN159129 |
| *D. mutila* | ALG101 | *Citrus*/wood canker | Algeria, Chiffa | Akila Berraf-Tebbal | MN104131 | MN159130 |
| *Dothiorella viticola* | ALG83 | *Citrus*/wood canker | Algeria, Staoueli | Akila Berraf-Tebbal | MN104087 | MN159086 |
| *Doth. viticola* | ALG35 | *Citrus*/wood canker | Algeria, Oued El Alleug | Akila Berraf-Tebbal | MN104088 | MN159187 |
| *Doth. viticola* | ALG84 | *Citrus*/wood canker | Algeria, Chiffa | Akila Berraf-Tebbal | MN104089 | MN159188 |
| *Doth. viticola* | ALG85 | *Citrus*/wood canker | Algeria, Staoueli | Akila Berraf-Tebbal | MN104090 | MN159189 |
| *Doth. viticola* | ALG86 | *Citrus*/wood canker | Algeria, Oued El Alleug | Akila Berraf-Tebbal | MN104091 | MN159190 |
| *Doth. viticola* | ALG87 | *Citrus*/wood canker | Algeria, Chiffa | Akila Berraf-Tebbal | MN104092 | MN159191 |
| *Doth. viticola* | ALG88 | *Citrus*/wood canker | Algeria, Chiffa | Akila Berraf-Tebbal | MN104093 | MN159192 |

detected by applying Tukey's honestly significant difference (HSD) test. The R v. 3.5.1 statistical software was used to perform the statistical analysis.

## Nomenclature

The electronic version of this article in Portable Document Format (PDF) in a work with an ISSN or ISBN will represent a published work according to the International Code of Nomenclature for algae, fungi, and plants, and hence the new names contained in the electronic publication of a PLOS article are effectively published under that Code from the electronic edition alone, so there is no longer any need to provide printed copies. In addition, new names contained in this work have been submitted to MycoBank from where they will be made available to the Global Names Index. The unique MycoBank number can be resolved and the associated information viewed through any standard web browser by appending the MycoBank number contained in this publication to the prefix http://www.mycobank.org/MB/. The online version of this work is archived and available from the following digital repositories: PubMed Central and LOCKSS.

## Results

### Disease symptoms

Citrus dieback was detected in all the orchards and regions investigated, with different degrees of intensity. Various external symptoms, including partial or complete dieback of the tree, branch and shoot cankers, abnormal growth of epicormic shoots; defoliation and leaf chlorosis were observed. Moreover, in certain orchards, bark cracking of the trunk and the branches was also noticeable (Fig 1).

The analysis of the 80 symptomatic sweet orange (*Citrus sinensis*) trees sampled to carry out the isolations, revealed the existence of 92 necrotic lesions in the trunks and the branches. They belonged to four types of wood alteration, including: wedge-shaped necrosis (WSN), that was the most prevalent lesion (n = 30) of the total samples collected. The brown central necrosis (BCN) (n = 26) was the second most prevalent lesion, followed by the black spots in the xylem (BS) (n = 24) and yellow soft wood rot (YSW) (n = 12).

### Fungal isolation and identification

Isolation carried out from ninety-two samples yielded a total of forty-seven fungal colonies belonging to Botryosphaeriaceae. On the basis of morphological characteristics, it was possible to distinguish three morphological groups according to colour and shape of conidia. Twenty-five isolates with brown sub-globose and striate conidia were grouped as *Lasiodiplodia*-like fungi. Fifteen isolates with brown oblong to ovoid conidia as *Diplodia*-like fungi. A further seven isolates with brown, ovoid thick walled and 1-septate conidia were considered as *Dothiorella*-like fungi. The identification of the isolates was confirmed by analysis of ITS and *tef1*-α sequences, which distinguished five separate species. The BLAST searches in GenBank showed 99–100% identity with reference sequences of representative isolates including that of the ex-type. The identified species were: *D. seriata* (10 isolates), *D. mutila* (5 isolates), *Doth. viticola* (7 isolates), *L. mediterranea* (10 isolates) and a *Lasiodiplodia* sp. (14 isolates) that could not be assigned to any of the currently known species.

### Phylogenetic analysis

Phylogenetic analysis was performed using ITS and *tef1*-α sequences. Fragments of approximately 500 and 300 bases were determined for ITS and *tef1*-α regions, respectively. The ML

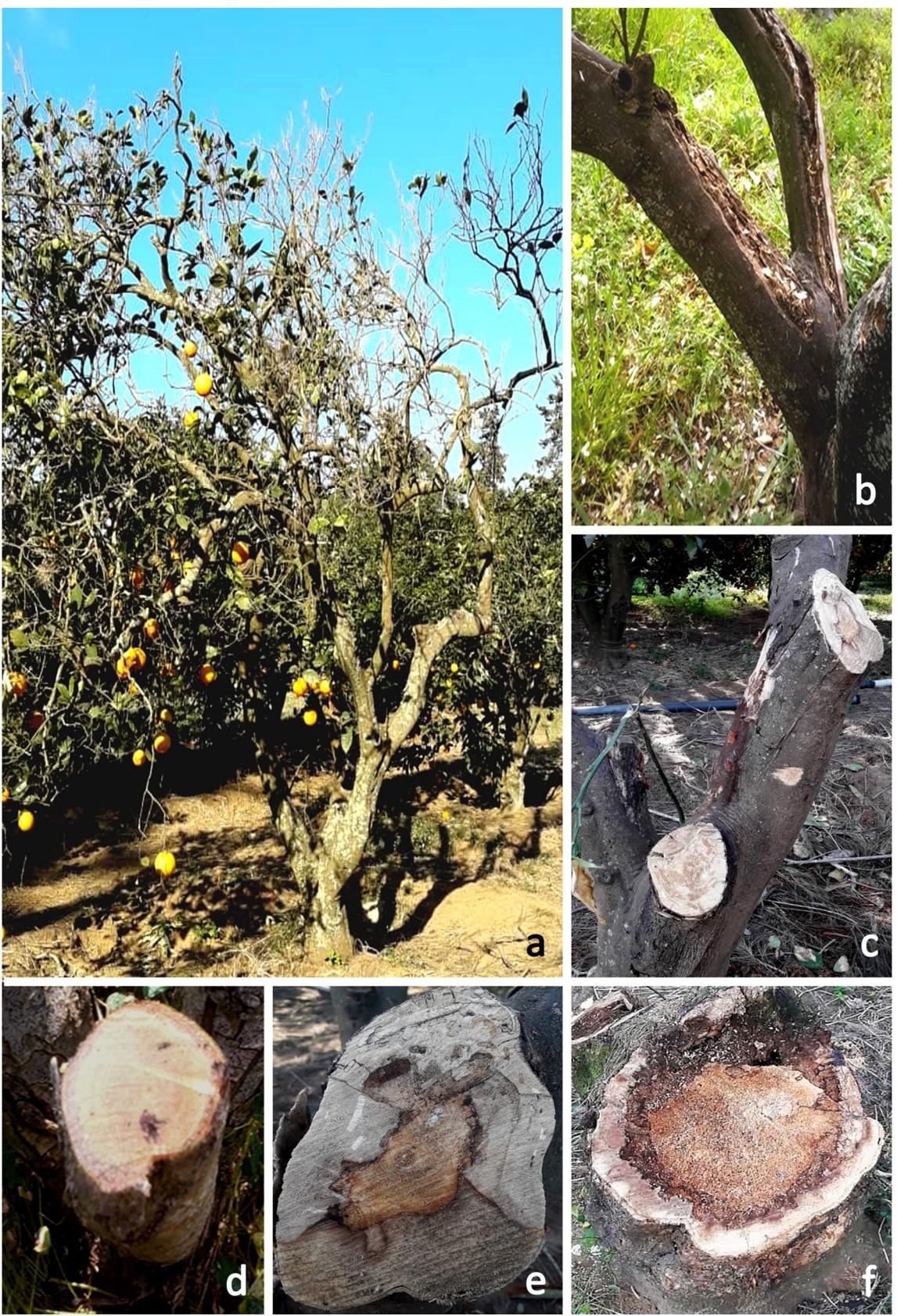

**Fig 1.** Citrus tree with dieback symptoms (a), bark cracking of the trunk and gummosis (b), main internal symptoms of sectioned branches and trunks (c–f).

and the MP trees are presented in Fig 2 and S1 Fig, respectively. The combined ITS and *tef1*-α dataset of *Lasiodiplodia* consisted of 23 isolates aligned with sequences of 69 isolates retrieved from GenBank, representing a selection of all known *Lasiodiplodia* and 2 outgroup taxa (*D. seriata* CBS 112555 and *D. mutila* CBS 112553). In the ML phylogenetic tree (Fig 2), the isolates obtained in this study grouped in two clades. The first clade comprised 10 isolates, which clustered together, with the ex-type strain of *L. mediterranea* (CBS 137783) and the ex-type strain of *L. vitis* (CBS 124060) (S1 Table), forming a single monophyletic group. The second group contained 14 isolates, which formed a distinct clade, with a high bootstrap support (ML/MP = 80/94), was considered to represent a distinct species, which is described here as *Lasiodiplodia mitidjana* sp. nov. (Fig 2).

## Taxonomy

*Lasiodiplodia mitidjana* A. Alves, A.E. Mahamedi & A. Berraf-Tebbal **sp. nov.** (Fig 3) [urn: lsid:mycobank.org:names: MB 832823]. Algeria, Mitidja, isolated from a branch canker of *Citrus sinensis*, June 2015, Akila Berraf-Tebbal, HOLOTYPE AVE-F-7, a dried culture sporulating on pine needles twigs deposited in the Herbarium Universitatis Aveirensis (AVE), culture ex-holotype MUM 19.90 (= ALG111). Other isolates examined are listed in Table 2.

**Etymology.** named after Mitidja where the fungus was discovered.

**Sexual state.** Not seen. Asexual state: Conidiomata stromatic, pycnidial, produced on pine needles on ¼ strength PDA within 2–3 wks, dark brown to black, covered with dense mycelium, superficial or immersed in the host becoming erumpent when mature, mostly uniloculate, solitary, globose, thick-walled. Paraphyses hyaline, cylindrical, thin-walled, initially aseptate, becoming septate when mature, rounded at apex. Conidiogenous cells holoblastic, discrete, hyaline, smooth, thin-walled, cylindrical, sometimes slightly swollen at the base. Conidia subovoid to ellipsoid-ovoid, apex rounded, occasionally tapering to truncate base, widest in middle to upper third, thick-walled, with granular content, initially hyaline and aseptate, remaining so for a long time, becoming dark brown and 1-septate, with longitudinal striations, (22.6–)27.7(−31.9) × (13.5–)16.7(−19.6) μm, 95% confidence limits = 27.3–28 × 16.5–16.9 μm (av. of 125 conidia ± SD = 27.7 ± 1.9 × 16.7 ± 1.1 μm, L/W ratio = 1.7).

**Cultural characteristics.** Colonies on PDA with moderate to dense aerial mycelium, initially white to smoke-grey, turning greenish grey on the surface and reverse, becoming dark slate blue with age.

**Cardinal temperatures for growth.** Minimum <10°C, maximum < 40°C and optimum 25–35°C, covering the medium surface (90 mm) before 7 days at 25°C in the dark.

**Habitat.** Twigs and branches of *Citrus sinensis*.

**Known geographic distribution.** Algeria.

**Notes.** Phylogenetically it is very closely related to *L. citricola* being distinguished by three bp in the *tef1*-α locus. Conidia tend to be larger than those of *L. citricola*, 95% confidence limits = 24.1–24.9 × 15–15.7 μm (av. ± S.D. = 24.5 ± 0.2 × 15.4 ± 1.8 μm) and have a lower L/W ratio = 1.6.

*Lasiodiplodia mediterranea* Linaldeddu, Deidda & Berraf-Tebbal **sp. nov.** (Linaldeddu et al. 2015. Fungal diversity 71:207)

**MycoBank.** MB 808356

**Synonym.** *Lasiodiplodia vitis* Yang & Crous, **sp. nov.** (Yang et al. 2017. Fungal Biology 121)

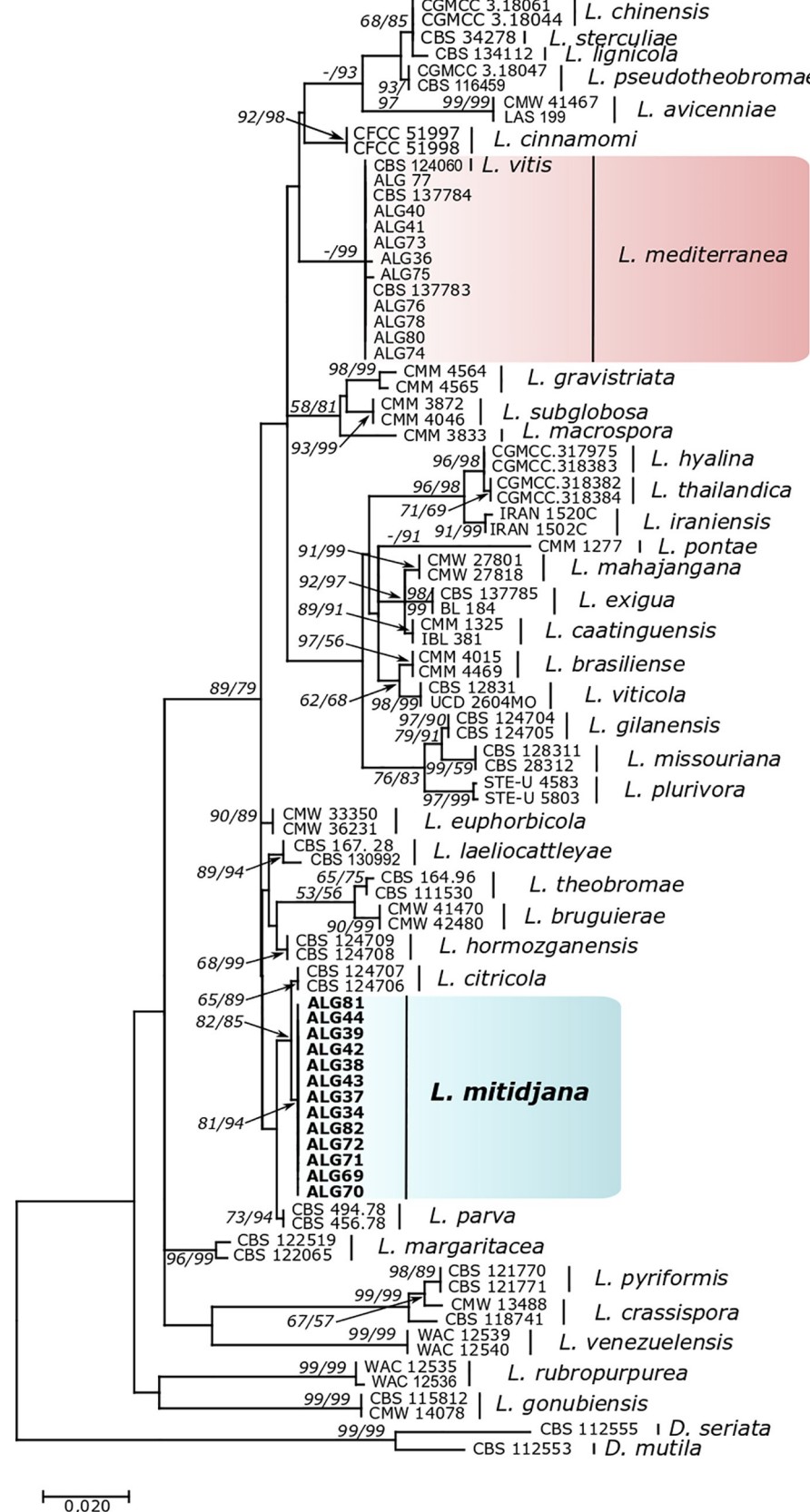

**Fig 2. Maximum likelihood tree generated from the combined analysis of ITS and *tef1*-α sequence data.** ML/MP bootstrap values are given at the nodes. Support values less than 50% are omitted or indicated with '–'. The tree was rooted to *D. mutila* and *D. seriata*.

**MycoBank.** MB817635

**Notes.** Yang et al. [40] described *L. vitis* as a novel species, clearly distinct from the species recognised on *Vitis vinifera* in Italy. However, in their study Yang et al. [40] did not include any representative of *L. mediterranea* which was described by Linaldeddu et al. [30] from several hosts, including *V. vinifera* in Italy. However, we have shown that *L. vitis* is phylogenetically indistinguishable from *L. mediterranea*. Their ITS sequences are 100% identical and the *L. vitis tef1*-α sequence deposited in GenBank differs from the *tef1*-α of *L. mediterranea* in 2 nt positions (1 missing G and a C instead of a T in the EF-986R primer binding region). We resequenced the *tef1*-α region of *L. vitis* CBS124060 (Table 2) using primers EF1-688F and EF1-1251R [35] which span a larger region than primers EF1-728F and EF-986R used by Yang et al. [40] and verified that these 2 nt are not real. There is no missing G in the *L. vitis* sequence and the C instead of a T is an artefact in *L. mediterranea* sequence introduced by the EF-986R primer sequence. Thus, the *tef1*-α region of *L. vitis* CBS124060 is 100% identical to the *tef1*-α sequence of *L. mediterranea*.

## Pathogenicity test

All the Botryosphaeriaceae isolates tested in the pathogenicity test were pathogenic to the citrus shoots. On the wood tissue under the bark, black to brown lesions developed, upward and downward from the inoculation point, within 30 days. The control plants did not develop any symptoms. Shapiro-Wilk test for normality revealed that the data differed significantly from a normal distribution (W = 0.83, p = 0.047). Lesion lengths varied between the species and among the isolates of each species tested, with a significant difference (F = 10.874; p < 0.001) (Table 3).

The most aggressive isolates were ALG91 (*D. seriata*) and ALG36 (*L. mediterranea*), which produced the longest lesions (5.49±2.65 cm and 4.39±1.31 cm, respectively) with a statistically significant difference recorded between ALG91 and the rest of the species, except for *L. mediterranea*. No significant difference in lesions size was observed between the isolates ALG40 (*L. mediterranea*) and ALG39 (*L. mitidjana*), which presented intermediate lesion lengths (3.83 ±0.97 and 3.88±1.24 cm, respectively). However, the smallest lesion size was produced by *Doth. viticola* ALG84 with 2.1±0.67 cm and both *D. mutila* isolates ALG102 (2.04±0.54) and ALG103 (2.05±0.4 cm). *D. seriata* was the only species that showed significant difference in lesion length between its two isolates (Table 3).

Koch's postulates were confirmed by a successful re-isolation of all tested fungal species from the necrotic tissues (Table 3).

## Distribution of Botryosphaeriaceae species

Overall, the Botryosphaeriaceae species occurred in 42 of the 80 citrus trees showing canker and dieback symptoms (S2 Table). Five distinct Botryosphaeriaceae species were obtained in this study. Each species was found with its respective frequency, as follow: *L. citricola* (14.1%), *L. mediterranea* (13%) and *D. seriata* (10.9%), *Doth. viticola* (7.6%) and *D. mutila* (5.4%).

At least, two different species were found in each orchard. *L. mediterranea* and *Doth. viticola* were found in six of the ten surveyed orchards. They were followed by *L. mitidjana*, recorded from five orchards of two municipalities. *D. seriata* was found in four sampling sites; whereas, *D. mutila* was recovered from only two orchards of the same municipality.

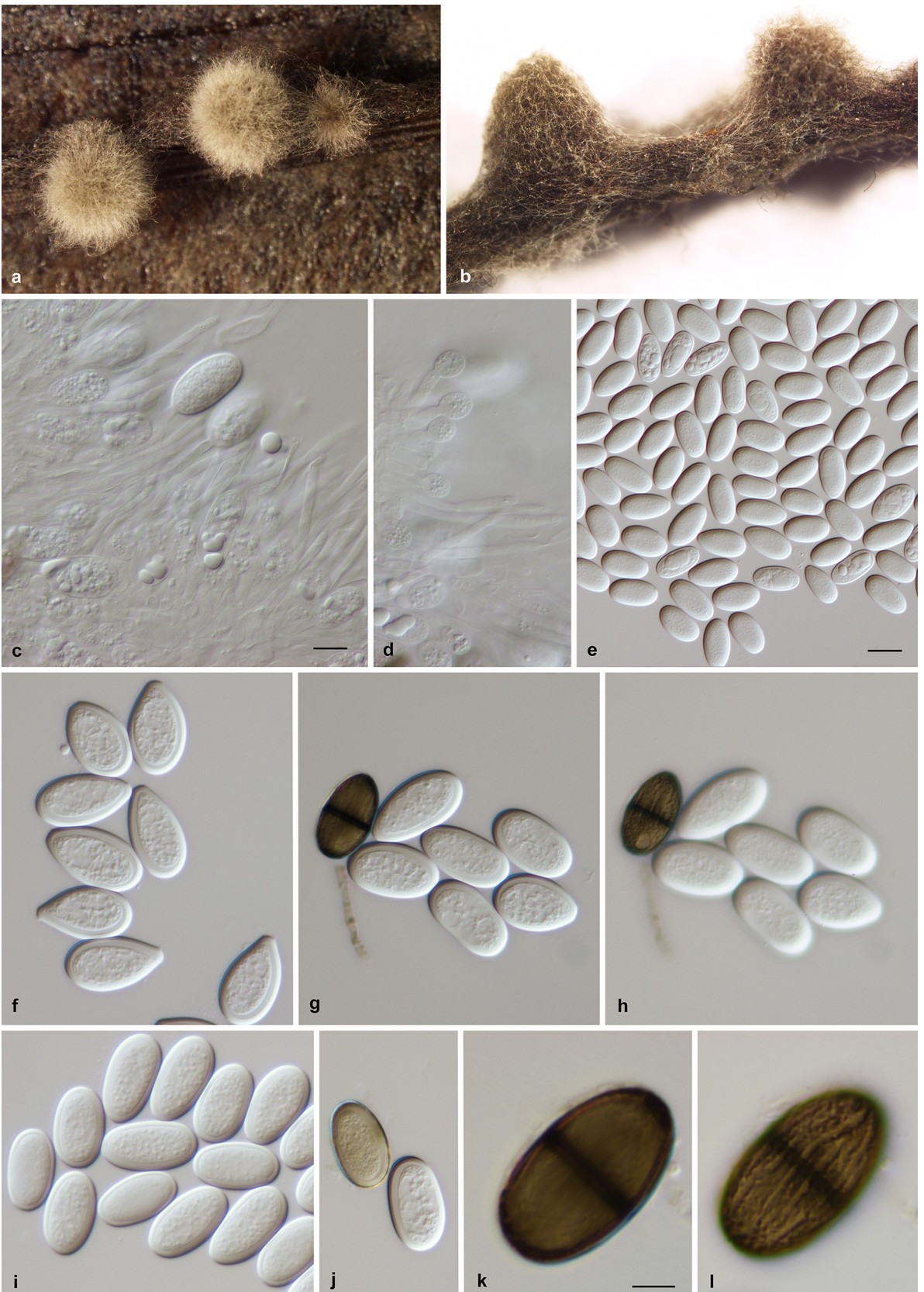

**Fig 3. Lasiodiplodia mitidjana.** (a,b). Pycnidia formed on pine needles. (c). Conidiogenous layer with conidia developing on conidiogenous cells. (d). Conidia developing on conidiogenous cells and paraphyses. (e,f,i). Hyaline aseptate conidia. (g,h). Hyaline aseptate brown 1-septate conidia in two focal planes showing the striations on the inner surface of the wall. (j). Aseptate conidia, one becoming brown. (k,l). Brown 1-septate conidia in two focal planes to show the striations in the inner surface of the wall. Scale bars: e = 20 μm, c,d,f–j = 10 μm; k–l = 5 μm.

## Discussion

This study aimed to evaluate and characterize the diversity of the Botryosphaeriaceae species associated with dieback of *Citrus sinensis*. It represents the first survey and preliminary investigation of these species in the main citrus orchards in northern Algeria.

Citrus canker and dieback were detected in all regions surveyed. Several external symptoms, including partial or complete dieback of the tree, branch and shoot cankers were observed. Over time, the disease can increase and seriously affected trees can become barren and eventually, die. Similar situations have been described in several citrus orchards, worldwide [8, 17, 22, 23, 41]. According to previous studies, abiotic factors, including drought, severe sunburn or freezing predispose the trees to xylem dysfunction, leading to these diseases [3, 4, 41].

In this study, five species belonging to three different genera of the Botryosphaeriaceae were recovered from symptomatic citrus trees, namely: *L. mediterranea*, *D. seriata*, *D. mutila*, *Doth. viticola* and *L. mitidjana*. The latter is introduced here, as new species. To our knowledge, except for *D. seriata* and *L. mediterranea*, this is the first report of *D. mutila*, *Doth. viticola* and *L. mitidjana*, causing branch canker disease on citrus and any crop, in Algeria. The Botryosphaeriaceae species were recovered from more than half of the trees sampled and were found in all the prospected orchards.

*Lasiodiplodia* was the most commonly isolated genus that was found in six of the ten surveyed orchards. This fact is consistent with previous studies, which showed that *Lasiodiplodia* species have the ability to target a wide variety of plants, distributed worldwide [42–44]. In fact, *Lasiodiplodia* species do not only occur as latent endophytes in asymptomatic plants but are also associated with different symptoms occurring on a variety of hosts including stem-end rot, fruit rot, decline, canker and dieback [43, 45–47]. In this study, *L. mediterranea* and *L. mitidjana* sp. nov. were the most frequently encountered species. *L. mediterranea* has been

**Table 3. Mean lesion lengths (cm) caused by *Doth. viticola*, *D. mutila*, *D. seriata*, *L. mediterranea* and *L. mitidjana* species implicated in citrus dieback in northern Algeria, 30 days after inoculation of detached green branches with mycelium-colonized agar plugs.**

| Species | Isolate code | Mean lesion length (cm) ± SD |
|---|---|---|
| *D. seriata* | ALG91 | 5.49±2.65 a |
| *D. seriata* | ALG98 | 2.35±0.55 cd |
| *L. mediterranea* | ALG36 | 4.39±1.31 ab |
| *L. mediterranea* | ALG40 | 3.83±0.97 abc |
| *Doth. viticola* | ALG86 | 2.82±0.86 bcd |
| *Doth. viticola* | ALG84 | 2.1±0.67 d |
| *D. mutila* | ALG102 | 2.04±0.54 d |
| *D. mutila* | ALG103 | 2.03±0.29 d |
| *L. mitidjana* | ALG39 | 3.88±1.24 abc |
| *L. mitidjana* | ALG34 | 2.2±0.67 cd |

The same letter after numbers refers to the isolates that do not differ significantly according to Tukey's HSD test at P ≤ 0.05.

reported as the causal agent of canker and dieback of grapevine, holm oak as well as citrus, indicating its capability to target different hosts [30]. The latter findings lead Andolfi et al. [48] to isolate and characterize the main secondary metabolites produced by *L. mediterranea*, as well as to evaluate its phytotoxic and antifungal activities. According to the former authors [30], *L. mediterranea* has been found only on V-shaped necrotic sectors of grapevine while, it has been isolated from all the lesion types of citrus trees in this study. *L. mitidjana* sp. nov. was found in five of the surveyed orchards. Isolates of this species were present predominantly in the wedge-shaped necrosis.

*Doth. viticola* was isolated at low frequency compared to *Lasiodiplodia* species found in this study. Interestingly, it was detected in six sampling sites. This species was first described as *Spencermartinisia viticola* by Phillips et al. [49]. It was obtained for the first time from *Vitis vinifera* in Spain. Recently, Yang et al. [40] regarded *Spencermartinsia* a synonym of *Dothiorella* and thus transferred the epithet *viticola* to *Dothiorella* as *Doth. viticola*. This taxonomic change was supported by a multi-gene phylogeny that included *Spencermartinsia* in *Dothiorella* genus [40, 50]. *Doth. viticola* has been reported from a wide range of woody hosts, including citrus trees [18, 23]. Recently, it has been also described as the causal agent of gummosis on citrus trees, in Tunisia [22]. According to Phillips et al. [20] and Dissanayake et al. [51], this species is known from China, Chile, USA, Spain, France, Australia, South Africa and Tunisia. Therefore, this study constitutes the first record of *Doth. viticola* in Algeria, which thus expands its known geographical range.

Two species of *Diplodia* genus, *D. seriata* and *D. mutila* were isolated from the surveyed orchards. *Diplodia* species are well known to cause damage on several economically important species and causing numerous disease symptoms including blight, dieback, rot diseases and canker [24, 30, 52–56]. In this study, *D. seriata* was frequently recovered from the sampling sites, which matches the findings of previous studies indicating the cosmopolitan nature of this species. This latter species is commonly reported as a pathogen on a large number of hosts and has been reported from hundreds of plant species [20, 51, 52]. *D. mutila*, the second *Diplodia* species isolated in this study, was less frequently found in the prospected orchards. Moreover, to our knowledge, this is the first report of this species in Algeria. In addition to Algeria, the USA is the only other country in which both *D. seriata* and *D. mutila* have been associated with citrus dieback [23]. These species have been found on apples in the USA [57, 58], Chile [59], France [60], Germany [61], Uruguay [62] and South Africa [63]; as well as in pear trees [62, 63], plum [64], peach and apricot [62, 65] and walnut [53].

All the Botryosphaeriaceae species of this study caused necrosis on the citrus shoots, with differences in the lengths of the lesions. These differences were observed between the species and also among isolates of the same species. According to the obtained results, the largest lesion was observed from one isolate of the *D. seriata* species (5.49 cm). Indeed, *D. seriata* was significantly different compared to the rest of the isolates, which is consistent with previous studies that showed significant impact of *D. seriata* on several hosts, across the globe [30, 66–68]. However, for this study, further tests need to be conducted in order to confirm our findings. According to the data from Table 3, considering the species aggressiveness, we could say that *L. mediterranea* is the most aggressive due to the size of both isolates tested in this study, with lesion lengths of 4.39 cm and 3.83 cm respectively.

For *L. mediterranea*, our results are in accordance with a previous study, which highlighted its aggressiveness in artificial inoculation experiments [30]. The smallest lesions were obtained from the isolates of *D. mutila* and *Doth. viticola*. Nevertheless, this was not the case in another study comparing *D. mutila* to *L. theobromae* and *D. seriata*, in which *D mutila* was found to have the largest lesions length [62]. According to Linaldeddu et al. [30] and Chakusary et al. [52], these differences in aggressiveness maybe due to several factors including genetic

variability of isolates, age, type of host tissue, differences in susceptibility as well as inoculation methods and experimental conditions. In this case, extensive sampling from citrus as well as other hosts are required to further emphasise the findings and draw a final solid conclusion.

Overall, almost all the Botryosphaeriaceae species we identified have previously been detected on citrus trees with the exception of *L. mitidjana*, which was described for the first time associated with citrus dieback. Given the major impact of the Botryosphaeriaceae species isolated on declining trees, worldwide, it is important to emphasize the urgent need to implement prevention techniques and management strategies in order to minimize the incidence of these pathogens and to prevent their spread to new orchards. For a better understanding of citrus dieback, it is necessary to set up larger surveys that include all citrus production areas. These surveys would assess, more accurately, the impact of the trunk diseases pathogens and eventually identify the factors that influence the dieback. This will be set in order to identify a number of practices to prevent their development.

## Supporting information

**S1 Fig. Maximum Parsimony phylogenetic tree resulting from the analysis of the combined ITS and *tef1*-α sequence data from *Lasiodiplodia* species.** The tree was rooted to *Diplodia mutila* and *Diplodia seriata*.
(TIF)

**S1 Table. Details of strains included in the phylogenetic and/or morphological analyses.**
(DOCX)

**S2 Table. Distribution of the Botryosphaeriaceae species among the surveyed orchards.**
(DOCX)

## Author Contributions

**Conceptualization:** Akila Berraf-Tebbal, Artur Alves.

**Data curation:** Akila Berraf-Tebbal, Alla Eddine Mahamedi.

**Formal analysis:** Akila Berraf-Tebbal, Artur Alves.

**Investigation:** Akila Berraf-Tebbal, Wassila Aigoun-Mouhous, Milan Špetík, Aleš Eichmeier.

**Methodology:** Akila Berraf-Tebbal, Alla Eddine Mahamedi, Artur Alves.

**Project administration:** Akila Berraf-Tebbal, Jana Čechová, Robert Pokluda, Miroslav Baránek, Aleš Eichmeier.

**Software:** Alla Eddine Mahamedi.

**Supervision:** Akila Berraf-Tebbal, Artur Alves.

**Writing – original draft:** Akila Berraf-Tebbal, Alla Eddine Mahamedi, Wassila Aigoun-Mouhous.

**Writing – review & editing:** Akila Berraf-Tebbal, Artur Alves.

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
