## [Decision Letter · Decision Letter 0]

4 Dec 2019

PONE-D-19-29810

*Lasiodiplodia mitidjana* sp. nov. and other Botryosphaeriaceae species causing branch canker and dieback of *Citrus sinensis* in Algeria

PLOS ONE

Dear Dr. BERRAF-TEBBAL,

Thank you for submitting your manuscript to PLOS ONE. After careful consideration, we feel that it has merit but does not fully meet PLOS ONE’s publication criteria as it currently stands. Therefore, we invite you to submit a revised version of the manuscript that addresses the points raised during the review process.

We have now obtained two thoughtful and very complete reports from external reviewers on study PONE-D-19-29810 “*Lasiodiplodia mitidjana* sp. nov. and other Botryosphaeriaceae species causing branch canker and dieback of *Citrus sinensis* in Algeria”. Both reviewers and myself agree that this work addresses interesting questions as still very little is known about Botryosphaeriaceae in Algeria.  The study is well presented and the reviewers are generally positive about the results, however, they have also identified some major concerns. Based on the reviews and my own assessment, I am recommending that the authors should resubmit a revised version of the paper that takes note of all the reviewers' comments. They should also include a detailed letter describing how the concerns have been taken into account. In their revision, the authors should not overstate what can be concluded from the results and, therefore, add caveats where necessary.

Please also verify that all the data generated for this study has been made available. Reviewer 1 mentioned that the GenBank accession for the Tef1-α sequences was not available online.

Some additional minor remarks:

L.273 ..we have shown…

L. 279 “…these 2 nt are not real.” Are not real is confusing, maybe instead use “ were correctly determined”?

L. 335 “The remaining species…” better mention the species’ name here.

L. 351 A similar situation has been…

L. 363 worLd

L. 373 compared instead of comparing

L. 389 “…known for targeting economically important plants…” targeting sounds as if they are selecting the hosts based on the economical value. Maybe better say: …known to cause damage on several economically important species...”

l. 393 “The later…”, better say “This latter species..” or just “It...”

L. 408 …with a previous study…

I would suggest to add some research perspectives at the end of the discussion. What kind of studies could help to better understand and develop management recommendations against citrus dieback?

We would appreciate receiving your revised manuscript by Jan 18 2020 11:59PM. To enhance the reproducibility of your results, we recommend that if applicable you deposit your laboratory protocols in protocols.io, where a protocol can be assigned its own identifier (DOI) such that it can be cited independently in the future. For instructions see: http://journals.plos.org/plosone/s/submission-guidelines#loc-laboratory-protocols

We look forward to receiving your revised manuscript.

Kind regards,

Katharina B Budde, Ph.D.

Academic Editor

PLOS ONE

Journal Requirements:

2. In your Methods section, please provide additional location information, including geographic coordinates for the data set if available.

3. Thank you for stating the following financial disclosure:"NO - Include this sentence at the end of your statement: The funders had no role in study design, data collection and analysis, decision to publish, or preparation of the manuscript."

Please provide an amended Funding Statement that declaresand names *all* the funding or sources of support received during this specific study (whether external or internal to your organization) as detailed online in our guide for authors at http://journals.plos.org/plosone/s/submit-now.  

Additional Editor Comments (if provided):

Reviewers' comments:

Reviewer's Responses to Questions

**Comments to the Author**

1. Is the manuscript technically sound, and do the data support the conclusions?

Reviewer #1: Partly

Reviewer #2: Yes

2. Has the statistical analysis been performed appropriately and rigorously? 

Reviewer #1: N/A

Reviewer #2: Yes

3. Have the authors made all data underlying the findings in their manuscript fully available?

Reviewer #1: No

Reviewer #2: Yes

4. Is the manuscript presented in an intelligible fashion and written in standard English?

Reviewer #1: Yes

Reviewer #2: Yes

5. Review Comments to the Author

Reviewer #1: General comments:

Akila Berraf-Tebbal and collaborators present in this manuscript the results of a survey on Botryosphaeriaceae diversity and pathogenicity affecting symptomatic sweet orange (Citrus sinensis) in Algeria. Botryosphaeriacea species were identified in every orchard tested (n=10) and a total of five different species have been isolated from symptomatic samples, with frequencies ranging from 5.4% to 14.1% of the samples. One species appears to form a distinct monophyletic group - not described before - and is claimed by the authors to be a new species. The pathogenicity of two representative isolates for every species isolated (n=5) was tested experimentally on Citrus shoot and Koch’s postulate was verified for all of them. Differences in pathogenicity was observed between isolates.

Overall, the manuscript is well written and there is no doubt about the efforts made by the authors to produce this work. The results presented are interesting and alert about the potential spread and threats brought by these pathogens, as mentioned by the authors.

My main comments to improve this manuscript is that authors often goes on conclusions that are not completely supported by the results. I suggest therefore to reconsider these conclusions or change the way they are presented to stick more on the facts. Furthermore, the GenBank accession for the Tef1-α sequences are not available which hamper my reviewing conclusions. This is even more problematic because the genetic difference observed to discriminate the new species are brought by this marker.

In this sense, a new species (Lasiodiplodia mitidja) is introduced in this study. This introduction is based on a two loci phylogeny, as well as morphological observations. I’m not a taxonomist myself, but are two SNPs (which I could not verified, and that is not illustrated by an alignment neither in the manuscript) and a bootstrap of 80 enough to consider the organism a new species? Concerning the morphological differences, as the authors mentioned, conidia “tend” to be larger and L/W ratio is different but for both measurements, no statistical significance is brought to the observations to confirm the difference. Can it represent a subpopulation of L. citricola? I presume it is not possible to test if those “species” outcross but if we have to be more rigorous, I would recommend to stay more prudent about the “new species” terminology and presented it more as a suggestion, or inform the readers that all the criteria to say it’s a new species are not completely fulfil.

My second concern is the ambiguity made between the types of wood alteration/symptoms observed in Citrus trees and the presence and implication of Botryosphaeriaceae. Botryosphaeriaceae species can be isolated from certain types of alteration and yet not being responsible of these alterations. Knowing the “opportunist” behaviour of these pathogens, I would not be surprised if they take over the habitat after a disequilibrium was induced into the microbiome of trees following another pathogen attack. The fact that the isolates were able to provoke symptoms experimentally does not necessarily mean they are responsible of the ones observed on the diseased trees, especially since the symptoms observed after artificial inoculation are not correlated to the ones observed on fields. Similarly, the presence of basidiocarps on heavily symptomatic Citrus is confusing for me, at least the way it is presented. What is the link between the Botryosphaeriaceae and the basidiocarps emergence, which species correspond to this basidiocarp? In a similar fashion, Figure 4 is confusing as my conclusion on this figure is that Botryosphaeriaceae can be isolated from different types of symptoms and not that one species is more isolated from one type of symptoms than another as the authors tend to say. There is no statistic proving so, and a quick interpretation (but false) from a hurried reader would be that such species is responsible of such symptoms. At this point, those results are more detrimental that beneficial to the study. I either recommend to delete this part or erase those ambiguities by a deeper discussion and a clearer result presentation. What do we know about the multifactorial aspects of dieback diseases? Is there only one pathogen involved? I think study conducted on Botryosphaeriaceae and grapevine trunk disease can be related to this case. Furthermore, if this part is conserved, more insights on what is known about the different symptoms that can cause Botryosphaeriaceae could/should be presented in the introduction.

Finally, the statistical methods used to test pathogenicity differences is either not well presented or the conclusions are not correct. This part needs to be improve. Have you tested species effect? Isolate effect? What are the p-value attributed to each ANOVA test, which factor has been tested by the ANOVA? Is LSD method (which is not described by the way) the more appropriate in your case?

Minor Comments:

L30: 14.1% percent of the samples and 13% of the samples

L31: what is the difference between widespread and abundant?

L42: I would erase (pomelos)

L43: Despite the high adaptation capacity of citrus trees to different climates (reference is missing)

L47: Citrus diseases are numerous and diverse, and are caused by phytopathogenic agents belonging to viruses, viroids, phytoplasmas, bacteria, and fungi (reference is missing).

L57: reference is missing

L63: colonize or affect?

This part on Botryosphaeriaceae should be more documented: classification of Botryosphaeriaceae, how many genera, endophytes with symptomless period, etc…

L82: Surveys were conducted in ten commercial orchards in the northern region of Algeria, specifically, in the Mitidja plain at the base of the Tell Atlas Mountains (Table 1).

Table 1: I would add coordinates of the orchards and years of sampling.

L94: was the scalpel sterilized?

L112: In this paragraph, can you add more infos about the PCR conditions?

L120-121: pyrosequencing?

L123: Newly generated sequences were deposited in GenBank (Table 2): the Tef1-α isn’t accessible.

L124: Sequences for both DNA regions were retrieved in BLAST searches from GenBank [34].

Check the meaning of this sentence. For example: “Homological sequences of the newly sequenced ones were retrieved from the GenBank by Blast.”

L125: Table 2: Can you add more info on this table, like type of tissue (trunk/branches), type of symptoms (under your classification), Orchard, etc….

L132: Please specify the request you made, or put the sequence in the supplemental files.

L136: what kind of adjustments?

L153: diameter

L164: this part need to be more precise/improve. Which threshold to accept the significance of the ANOVA, which factor tested, what LSD means for, which soft did you use?

L171: nothing is said about the distribution

L172-173 and L174-175 could be fused for clarity purposes

L175: the total number of samples… Samples = branches?

L176 – 177: this measurement is completely arbitrary and according to me abusive. If one pathogen would have occurred at 80% frequency, the difference between 11%(very frequent) and 4%(infrequent) would be meaningless. My advice, stick to the numbers and do not try to interpreted it in a frequent or infrequent way, that’s too subjective.

L186: On heavily infected trees, basidiocarps emerged: that’s ambiguous, as said before.

L189: why wedge-shaped necrosis is not name WSN?

L191: similar BCN instead of NBC and YSW instead of NCC.

L196: the “e.” is missing on the picture, but maybe see in it as a sign for not putting this picture…

L214: Why MP tree is not shown? At least in supplemental file?

L218: why 23 isolates and not 24 (10+14)

L228: The phylogenetic tree of only one method is presented, although the bootstrap values of the two methods are shown. Can we see the tree constructed with the second method in the supplemental files?

L279: can you have the sequence accessible please?

L311: from what I’ve read, LSD test is not recommended anymore as sensitive to multiple comparison. Furthermore, as mentioned above, this test is not well conducted and presented. A histogram would be better, with a sign for significant difference, either at the isolate level and species level, with threshold use for significance. The 100% of re-isolation frequency for every isolates are not necessary in the table according to me, if you say it in the text.

L317: Ambiguous: are you speaking of distribution in the wood? If yes, cut this paragraph in two: Frequency of occurrence / Distribution of Botryosphaeriaceae in the wood.

L320: I comment already the frequent and very frequent ranking, that’s abusive according to me.

L323: For each orchard, at least two different species were isolated, average per orchard?

L328: new paragraph or no paragraph at all as mentioned above, I found this part confusing.

L370-371: According to these authors, L. mediterranea has been found only on V-shaped necrotic sectors of grapevine while it has been isolated from all the lesion types of citrus trees in this study.

L373: what the results of Andolfi et al. bring to your results?

L374-375: maybe if you had isolated only the 4 or 5 isolates that were from brown central necrosis, would you have said that the species was exclusively found in brown central necrosis? This part of the discussion is not really constructive.

L387: More interesting that this, it confirms its wide geographical range.

L402: Wedge shaped lesion?

L403-405: this part of the discussion goes beyond what your data show, either improve your statistical analysis or moderate the message.

L408-415: similar, hard to have this kind of discussion with two isolates per strains, with one phenotyping trial.

L446: References have to be reformatted: species not in italic, capital letter on every first word letter for some references, etc.; some other problem like L508 Phililips is written bizarrely.

Reviewer #2: This is a nice study of Botryosphaeriaceae which are important plant pathogens, including fruit trees. Algeria is unexplored both from mycological and pathological perspective and it is nice to see collaborations like this resulting in a good piece of work. I agree with authors regarding Lasiodiplodia mediterranea and L.vitis situation, especially because pcr atifacts introduced by primer sequences are unfortunately a common thing these days (my personal experience). I have small suggestions that would improve the paper.

Abstract, line 25-of Botryosphaeriaeceae

Abstract, line 29- Delete which, add Lasiodiplodia mithidjana is described in this paper as a new species

Abstract, line 62- delete effect

Materials-line 98-dried on sterilized paper (towels, filter paper?)

Materials, line 100-The mycelium emerging from wood pieces was transferred…

Materials, line 105-Isolates that lacked pycnidia production…

Materials, line 150-How did you select representative isolates?

Materials, line 151-Shoots? But above you mentioned branches (line 148)

Materials, line 157-“… well watered and maintained under favorable conditions” What do you mean by this? Were the cuttings in soil or in water? How many times per week did you change the water? Ambient temperature? Light?

Results, line 182-Degrees of intensity? Where are they?

Results, line 186-Basidiocarps of which species or genera?

Results, line 309-What about control plants?

Results, line 313-Now you mention branches again

Discussion, line 360-and seriously affected trees can become

Discussion, lines 403-404- “However, D. seriata was significantly different compared to the rest of the isolates” But previously you said that both D. seriata and L. mediterranea were most aggressive species (based on lesion lengths). So which species was in fact most aggressive? Also, what about differences in aggressiveness between different isolates of the same species?

Discussion, lines 408-412-Was this previous study also about Bot on citrus trees?

Discussion, lines 409 and 411-In line 404 you are talking about aggressiveness. Now about virulence. In line 413 you talk again about aggressiveness. Virulence and aggressiveness don’t mean the same thing. Replace the term virulence with aggressiveness in lines 409 and 411.

6. PLOS authors have the option to publish the peer review history of their article (what does this mean?). If published, this will include your full peer review and any attached files.

Reviewer #1: Yes: Benoit LAURENT

Reviewer #2: No

---

## [Author Response · Author response to Decision Letter 0]

19 Jan 2020

Answers to the reviewer 1

General comments:

Akila Berraf-Tebbal and collaborators present in this manuscript the results of a survey on Botryosphaeriaceae diversity and pathogenicity affecting symptomatic sweet orange (Citrus sinensis) in Algeria. Botryosphaeriacea species were identified in every orchard tested (n=10) and a total of five different species have been isolated from symptomatic samples, with frequencies ranging from 5.4% to 14.1% of the samples. One species appears to form a distinct monophyletic group - not described before - and is claimed by the authors to be a new species. The pathogenicity of two representative isolates for every species isolated (n=5) was tested experimentally on Citrus shoot and Koch’s postulate was verified for all of them. Differences in pathogenicity was observed between isolates.

Overall, the manuscript is well written and there is no doubt about the efforts made by the authors to produce this work. The results presented are interesting and alert about the potential spread and threats brought by these pathogens, as mentioned by the authors.

My main comments to improve this manuscript is that authors often goes on conclusions that are not completely supported by the results. I suggest therefore to reconsider these conclusions or change the way they are presented to stick more on the facts. 

We took into consideration all the comments regarding the conclusions and we made sure to do all the required modifications.

Furthermore, the GenBank accession for the Tef1-α sequences are not available which hamper my reviewing conclusions. This is even more problematic because the genetic difference observed to discriminate the new species are brought by this marker.

The ITS and the tef1-α sequences have been deposited into GenBank; However, the tef1-α sequences are not automatically deposited into GenBank after being accessioned. Each sequence record is individually examined and processed by the GenBank annotation staff to ensure that it is free of errors or problems.

In this sense, a new species (Lasiodiplodia mitidja) is introduced in this study. This introduction is based on a two loci phylogeny, as well as morphological observations. I’m not a taxonomist myself, but are two SNPs (which I could not verified, and that is not illustrated by an alignment neither in the manuscript) and a bootstrap of 80 enough to consider the organism a new species? Concerning the morphological differences, as the authors mentioned, conidia “tend” to be larger and L/W ratio is different but for both measurements, no statistical significance is brought to the observations to confirm the difference. Can it represent a subpopulation of L. citricola? I presume it is not possible to test if those “species” outcross but if we have to be more rigorous, I would recommend to stay more prudent about the “new species” terminology and presented it more as a suggestion, or inform the readers that all the criteria to say it’s a new species are not completely fulfil.

We agree that this may be debatable. In fact, we have discussed this previously within the team. However, it is clearly aligned with the current trend for introduction of novel Lasiodiplodia species.

In the future it may well be proven that in fact it is not a new species different from L. citricola. But taxonomy is dynamic and hence frequently changing. For the moment we would like to introduce the new species. The fact is that eventually it will be described as a new species, if not by our group, then by someone else.

As an example of a case which is similar to ours, here are the nucleotide differences for the following mentioned species: L. chinensis vs L. lignicola vs L. pseudotheobromae. For all 3 species ITS is 100 % identical 

As for Tef1:

L. chinensis vs L. lignicola: 1 nucleotide difference

L. chinensis vs L. pseudotheobromae: 3 nucleotides differences

My second concern is the ambiguity made between the types of wood alteration/symptoms observed in Citrus trees and the presence and implication of Botryosphaeriaceae. Botryosphaeriaceae species can be isolated from certain types of alteration and yet not being responsible of these alterations. Knowing the “opportunist” behaviour of these pathogens, I would not be surprised if they take over the habitat after a disequilibrium was induced into the microbiome of trees following another pathogen attack. 

We agree with the reviewer. After considering your other comments on the same part and your suggestion of removing it and given that it is not relevant for the paper we decided to delete it.

The fact that the isolates were able to provoke symptoms experimentally does not necessarily mean they are responsible of the ones observed on the diseased trees, especially since the symptoms observed after artificial inoculation are not correlated to the ones observed on fields.

We agree with the reviewer, however the goal was to test pathogenicity of the isolates and this is the way to do it. Of course we cannot be sure that they will behave the same way in the field but they have the potential to do so.

 Similarly, the presence of basidiocarps on heavily symptomatic Citrus is confusing for me, at least the way it is presented. What is the link between the Botryosphaeriaceae and the basidiocarps emergence, which species correspond to this basidiocarp? 

We described the health status of the orchards where the sampling has been carried out (branch and shoot cankers, abnormal growth of epicormic shoots; defoliation and leaf chlorosis). Basidiocarps are the fruiting bodies of the decay-causing fungi. Their presence on the trunk means that it is an already rotting trunk and that some ascomycetes (Botryosphaeriaceae, Diatrypaceae…) and Basidiomycetes (Fomitiporia, Phellinus….) have already colonized the trunk. 

In a similar fashion, Figure 4 is confusing as my conclusion on this figure is that Botryosphaeriaceae can be isolated from different types of symptoms and not that one species is more isolated from one type of symptoms than another as the authors tend to say. There is no statistic proving so, and a quick interpretation (but false) from a hurried reader would be that such species is responsible of such symptoms. At this point, those results are more detrimental that beneficial to the study. I either recommend to delete this part or erase those ambiguities by a deeper discussion and a clearer result presentation. What do we know about the multifactorial aspects of dieback diseases? Is there only one pathogen involved? I think study conducted on Botryosphaeriaceae and grapevine trunk disease can be related to this case. Furthermore, if this part is conserved, more insights on what is known about the different symptoms that can cause Botryosphaeriaceae could/should be presented in the introduction. 

We agree with the reviewer and his comments. After pointing out these remarks we thoughtfully considered them and decided to delete the figure 4 as well as the paragraphs related to it. 

Finally, the statistical methods used to test pathogenicity differences is either not well presented or the conclusions are not correct. This part needs to be improve. Have you tested species effect? Isolate effect? What are the p-value attributed to each ANOVA test, which factor has been tested by the ANOVA? Is LSD method (which is not described by the way) the more appropriate in your case? 

We took into consideration your valuable comments and we made sure to change this part and we removed all the ambiguities. 

Minor Comments:

L30: 14.1% percent of the samples and 13% of the samples

R: Revised as recommended

L31: what is the difference between widespread and abundant?

R: Widespread means that it is found or distributed over a large area. However, abundant means that it is existing or available in large quantities (it could have the same meaning as plentiful)

L42: I would erase (pomelos)

R: Revised as recommended

L43: Despite the high adaptation capacity of citrus trees to different climates (reference is missing)

R: Revised as recommended

L47: Citrus diseases are numerous and diverse, and are caused by phytopathogenic agents belonging to viruses, viroids, phytoplasmas, bacteria, and fungi (reference is missing).

R: Revised as recommended

L57: reference is missing

R: Revised as recommended

L63: colonize or affect?

R: We deleted ‘affect’

This part on Botryosphaeriaceae should be more documented: classification of Botryosphaeriaceae, how many genera, endophytes with symptomless period, etc…

R: Revised as recommended

L82: Surveys were conducted in ten commercial orchards in the northern region of Algeria, specifically, in the Mitidja plain at the base of the Tell Atlas Mountains (Table 1).

Table 1: I would add coordinates of the orchards and years of sampling.

R: Revised as recommended

L94: was the scalpel sterilized?

R: Yes, the scalpel was sterilized. This detail has been added to the manuscript.

L112: In this paragraph, can you add more infos about the PCR conditions?

R: Revised as recommended

L120-121: pyrosequencing?

R: the company used Sanger sequencing method.

L123: Newly generated sequences were deposited in GenBank (Table 2): the Tef1-α isn’t accessible.

R: The sequences are available in GenBank. 

L124: Sequences for both DNA regions were retrieved in BLAST searches from GenBank [34].

Check the meaning of this sentence. For example: “Homological sequences of the newly sequenced ones were retrieved from the GenBank by Blast.”

R: Revised as recommended

L125: Table 2: Can you add more info on this table, like type of tissue (trunk/branches), type of symptoms (under your classification), Orchard, etc….

R: Revised as recommended

L132: Please specify the request you made, or put the sequence in the supplemental files.

R: Revised as recommended (the sequences are in the supplementary files)

L136: what kind of adjustments?

R: The ITS and tef1-α sequences were initially aligned separately using ClustalX v. 1.83. The alignments were manually optimized by coding the missing sequences as “?”. Ambiguous sequences at the start and the end were deleted and gaps were adjusted in BioEdit. 

L153: diameter

R: Revised as recommended

L164: this part need to be more precise/improve. Which threshold to accept the significance of the ANOVA, which factor tested, what LSD means for, which soft did you use?

R: Revised as recommended

Detailed responses: 

Which threshold to accept the significance of the ANOVA

R: when the P value is below the threshold (0.05), the difference between the means is considered as significant.

Which factor tested?

R: We tested the lesions produced by each fungal isolate of the different species.

What LSD means for?

R: We changed the statistical test by using Tukey's honestly significant difference (HSD) test.

Which soft did you use?

R: The R v. 3.5.1 statistical software was used to perform the statistical analysis.

L171: nothing is said about the distribution

R: We removed the word ‘distribution’ from the title

L172-173 and L174-175 could be fused for clarity purposes

R: We removed this paragraph as recommended.

L175: the total number of samples… Samples = branches?

R: The samples mean the different necrotic lesions found in the branches and the trunks of the 80 trees.

L176 – 177: this measurement is completely arbitrary and according to me abusive. If one pathogen would have occurred at 80% frequency, the difference between 11%(very frequent) and 4%(infrequent) would be meaningless. My advice, stick to the numbers and do not try to interpreted it in a frequent or infrequent way, that’s too subjective.

R: We removed the paragraph related to the frequency of occurrence, as recommended.

L186: On heavily infected trees, basidiocarps emerged: that’s ambiguous, as said before.

R: We removed the description of the basidiocarps from the photoplate as well as from the text, as recommended. 

L189: why wedge-shaped necrosis is not name WSN?

R: Revised as recommended

L191: similar BCN instead of BCN and YSW instead of NCC.

R: Revised as recommended

L196: the “e.” is missing on the picture, but maybe see in it as a sign for not putting this picture…

R: Revised as recommended

L214: Why MP tree is not shown? At least in supplemental file?

R: Revised as recommended (the tree is in supplementary file)

L218: why 23 isolates and not 24 (10+14)

R: The sequence of one isolate was not good enough to use it for the phylogenetic analysis.

L228: The phylogenetic tree of only one method is presented, although the bootstrap values of the two methods are shown. Can we see the tree constructed with the second method in the supplemental files?

R: Revised as recommended 

L279: can you have the sequence accessible please?

R: The sequence has been submitted to GenBank. It will be available online after verification of the annotation. We have included the accession number into the table 2. 

L311: from what I’ve read, LSD test is not recommended anymore as sensitive to multiple comparison. Furthermore, as mentioned above, this test is not well conducted and presented. A histogram would be better, with a sign for significant difference, either at the isolate level and species level, with threshold use for significance. The 100% of re-isolation frequency for every isolates are not necessary in the table according to me, if you say it in the text.

R: Revised as recommended. 

L317: Ambiguous: are you speaking of distribution in the wood? If yes, cut this paragraph in two: Frequency of occurrence / Distribution of Botryosphaeriaceae in the wood.

R: Revised as recommended

L320: I comment already the frequent and very frequent ranking, that’s abusive according to me.

R: We deleted this paragraph. 

L323: For each orchard, at least two different species were isolated, average per orchard?

R: In the paragraph L323, we described the widespread of the species in the orchards. It was only to mention their presence on each orchard. 

For more details, here is a table containing all the information about species distribution among the surveyed orchards. 

 Region 

 Oued El Alleug Chiffa Boufarik Staoueli 

Species/ Orchards 1 2 3 4 5 6 7 8 9 10 Total 

D. seriata - - - - 2 1 - - 3 4 10

D. mutila - - - - 3 2 - - - - 5

L. mediterr. 2 2 3 1 - - 2 2 - - 12

L. mitidjana 0 3 3 3 - - 2 2 - - 13

Doth. vitic. 1 - 1 - 1 2 - - 1 1 7

Total 3 5 7 4 6 5 4 4 4 5 47

L328: new paragraph or no paragraph at all as mentioned above, I found this part confusing.

R: Revised as recommended (We removed the paragraph).

L370-371: According to these authors, L. mediterranea has been found only on V-shaped necrotic sectors of grapevine while it has been isolated from all the lesion types of citrus trees in this study.

R: Revised as recommended

L373: what the results of Andolfi et al. bring to your results?

R: Andolfi et al. (2016) isolated and characterized the main secondary metabolites produced by L. mediterranea. They also, evaluated its phytotoxic and antifungal activities. These findings support our results, which show the ability of this fungal species to colonize and cause damages in the wood.

L374-375: maybe if you had isolated only the 4 or 5 isolates that were from brown central necrosis, would you have said that the species was exclusively found in brown central necrosis? This part of the discussion is not really constructive. 

R: we removed the paragraph, as well as the figure 4.

L387: More interesting that this, it confirms its wide geographical range.

R: revised as recommended

L402: Wedge shaped lesion?

R: The paragraph is about the pathogenicity trial and the lesions produced by each Botryosphaeriaceae species. We did not consider the shape of the lesions, for the pathogenicity test.

L403-405: this part of the discussion goes beyond what your data show, either improve your statistical analysis or moderate the message.

R: Revised as recommended

L408-415: similar, hard to have this kind of discussion with two isolates per strains, with one phenotyping trial.

R: Revised as recommended

L446: References have to be reformatted: species not in italic, capital letter on every first word letter for some references, etc.; some other problem like L508 Phililips is written bizarrely.

R: Revised as recommended

Answers to the reviewer 2

Reviewer #2: This is a nice study of Botryosphaeriaceae which are important plant pathogens, including fruit trees. Algeria is unexplored both from mycological and pathological perspective and it is nice to see collaborations like this resulting in a good piece of work. I agree with authors regarding Lasiodiplodia mediterranea and L.vitis situation, especially because pcr atifacts introduced by primer sequences are unfortunately a common thing these days (my personal experience). I have small suggestions that would improve the paper.

Abstract, line 25-of Botryosphaeriaeceae

R: Revised as recommended

Abstract, line 29- Delete which, add Lasiodiplodia mitidjana is described in this paper as a new species

R: Revised as recommended

Abstract, line 62- delete effect

R: Revised as recommended

Materials-line 98-dried on sterilized paper (towels, filter paper?)

R: Revised as recommended

Materials, line 100-The mycelium emerging from wood pieces was transferred…

R: Revised as recommended

Materials, line 105-Isolates that lacked pycnidia production…

R: Revised as recommended

Materials, line 150-How did you select representative isolates?

R: We selected two isolates, from each phylogenetically resolved species.

Materials, line 151-Shoots? But above you mentioned branches (line 148)

R: Revised as recommended (We have standardized using shoot instead of branch).

Materials, line 157-“… well watered and maintained under favorable conditions” What do you mean by this? Were the cuttings in soil or in water? How many times per week did you change the water? Ambient temperature? Light?

R: The inoculated cuttings were wrapped with wet sterile cotton to avoid the desiccation of the agar plug. The shoots were immediately transplanted into pots containing sterilized water as a growth substrate (10 shoots per pot), which were incubated at the ambient room temperature, under daily photoperiod. The water of the container was changed twice a week.

Results, line 182-Degrees of intensity? Where are they?

R: The degrees of intensity refer to the different levels of the dieback symptoms observed in the orchards.

Results, line 186-Basidiocarps of which species or genera?

R: We did not identify the basidiocarps. We described all the symptoms related to the citrus trees dieback, including the fruiting bodies emerged from the trunks.

Results, line 309-What about control plants?

R: We did not isolate any of the tested species from the negative control.

Results, line 313-Now you mention branches again

R: revised as recommended

Discussion, line 360-and seriously affected trees can become

R: revised as recommended

Discussion, lines 403-404- “However, D. seriata was significantly different compared to the rest of the isolates” But previously you said that both D. seriata and L. mediterranea were most aggressive species (based on lesion lengths). So which species was in fact most aggressive? Also, what about differences in aggressiveness between different isolates of the same species?

So which species was in fact most aggressive?

R: The significant difference was made based on a comparison between all the tested isolates. It was not about the pairwise comparisons that take one isolate and compare it with each of the rest of isolates. D. seriata and L. mediterranea were the most aggressive species when compared to the rest of the species. However, D. seriata was the most aggressive species, considering the length of the lesion for each isolate, separately.

Also, what about differences in aggressiveness between different isolates of the same species?

R: Significant variation in aggressiveness can occur within and among isolates from the same species. This aggressiveness refers to the quantitative variation of pathogenicity on the susceptible host infection efficiency, the latent period, the spore production rate and the infectious period of each strain. These components are closely related to the genetic variability within the strains of the same species. 

Discussion, lines 408-412-Was this previous study also about Bot on citrus trees?

R: the study was about Lasiodiplodia species (Botryosphaeriaceae) on grapevine.

Discussion, lines 409 and 411-In line 404 you are talking about aggressiveness. Now about virulence. In line 413 you talk again about aggressiveness. Virulence and aggressiveness don’t mean the same thing. Replace the term virulence with aggressiveness in lines 409 and 411.

R: Revised as recommended 

Answers to the academic editor

L.273 ..we have shown…

R: Revised as recommended 

L. 279 “…these 2 nt are not real.” Are not real is confusing, maybe instead use “ were correctly determined”?

R: Revised as recommended 

L. 335 “The remaining species…” better mention the species’ name here.

R: Revised as recommended 

L. 351 A similar situation has been…

R: Revised as recommended 

L. 363 worLd

R: Revised as recommended 

L. 373 compared instead of comparing

R: Revised as recommended 

L. 389 “…known for targeting economically important plants…” targeting sounds as if they are selecting the hosts based on the economical value. Maybe better say: …known to cause damage on several economically important species...”

R: Revised as recommended 

l. 393 “The later…”, better say “This latter species..” or just “It...”

R: Revised as recommended 

L. 408 …with a previous study…

R: Revised as recommended 

I would suggest to add some research perspectives at the end of the discussion. What kind of studies could help to better understand and develop management recommendations against citrus dieback?

R: Revised as recommended

---

## [Decision Letter · Decision Letter 1]

17 Mar 2020

PONE-D-19-29810R1

Lasiodiplodia mitidjana sp. nov. and other Botryosphaeriaceae species causing branch canker and dieback of Citrus sinensis in Algeria

PLOS ONE

Dear Dr. BERRAF-TEBBAL,

Thank you for submitting your manuscript to PLOS ONE. After careful consideration, we feel that it has merit but does not fully meet PLOS ONE’s publication criteria as it currently stands. Therefore, we invite you to submit a revised version of the manuscript that addresses the points raised during the review process.

We have now received positive evaluations of manuscript PONE-D-19-29810 by two thoughtful reviewers who still ask for some mostly minor changes. I have read and evaluated the manuscript myself and agree with the reviewers. In the following, I will also provide some comments that might help to further improve the manuscript and make it more attractive to a broader readership.

I recommend starting out the introduction with a wider scope. Citrus cultivation is not only important in Algeria but in many countries and the fruits are being exported and eaten in even more countries, therefore I would start out with some sentences about the importance of Citrus in general. This could make the publication more attractive to a wider readership.

The authors mention at the end of the introduction that Botryosphaeriaceae have not been studied in detail in Algeria. Have other pathogen families or groups on Citrus already been studied in more detail? I think this should be introduced here in detail! This could also explain why the authors only focus on Botryosphaeriaceae and not check which other pathogens are causing the symptoms in their samples. Please explain that in more detail!

The objectives at the end of the introduction should be described in more detail! The authors do not even mention that they also aimed at testing the aggressiveness of the the isolates!

Table 1, I agree with Benoit Laurent that it would be relevant (if possible) to add the GPS coordinates of each orchard in each region as it would give an idea of how far apart the orchards are. If this information cannot be given, I would rather remove the GPS coordinates of the cities.

l. 145 Mention which was the best fitting DNA evolution model.

Please improve the description of the statistical analyses and test specifically for differences in aggressiveness between species as suggested by Benoit Laurent.

I am a little surprised about the paragraph called “Nomenclature”. I am not familiar with the description of new fungal species but I have never seen such a justification. Is this typical? Otherwise, it should be removed. I suggest to name this paragraph “Taxonomy” which should be mentioned that it has been described in detail and that the new names have been submitted to MycoBank.

In the results the authors mention that they obtained 47 fungal colonies belonging to Botryosphaeriaceae. Did they also observe other fungi in their isolates? What happened to the other samples?

l. 41 clementines

l. 62-64 This sentence is unclear. Do the authors refer to a single species of the Botryosphaeriaceae family or several that are all important pathogens? Please clarify!

l. 77 …citrus trees in Algeria…

l. 350 …similar situations have been described…

l. 351 …according to some authors… is not a good formulation!

We would appreciate receiving your revised manuscript by May 01 2020 11:59PM. To enhance the reproducibility of your results, we recommend that if applicable you deposit your laboratory protocols in protocols.io, where a protocol can be assigned its own identifier (DOI) such that it can be cited independently in the future. For instructions see: http://journals.plos.org/plosone/s/submission-guidelines#loc-laboratory-protocols

We look forward to receiving your revised manuscript.

Kind regards,

Katharina B Budde, Ph.D.

Academic Editor

PLOS ONE

Reviewers' comments:

Reviewer's Responses to Questions

**Comments to the Author**

1. If the authors have adequately addressed your comments raised in a previous round of review and you feel that this manuscript is now acceptable for publication, you may indicate that here to bypass the “Comments to the Author” section, enter your conflict of interest statement in the “Confidential to Editor” section, and submit your "Accept" recommendation.

Reviewer #1: (No Response)

Reviewer #3: All comments have been addressed

2. Is the manuscript technically sound, and do the data support the conclusions?

Reviewer #1: Partly

Reviewer #3: Yes

3. Has the statistical analysis been performed appropriately and rigorously? 

Reviewer #1: Yes

Reviewer #3: Yes

4. Have the authors made all data underlying the findings in their manuscript fully available?

Reviewer #1: Yes

Reviewer #3: Yes

5. Is the manuscript presented in an intelligible fashion and written in standard English?

Reviewer #1: Yes

Reviewer #3: Yes

6. Review Comments to the Author

Reviewer #1: I would like to thank Akila Berraf-Tebbal and his collaborators to the answers they brought to my questions and comments. The data on the tef1 alpha are now available, which was one major concern of my last review, and allow me to verify the monophyletic group corresponding to the introduction of a new species. The debate on this introduction is still questionable but does not question the quality and the presentation of the work. I don’t think that following a trend is really constructive and yet scientific, but I endorse their point of view about the dynamicity of the taxonomy. Besides, I’m rather surprised that, in response to my comment about the lack of general information on Botryosphaeriaceae, a sentence is brought from a reference and citing “It comprises 24 genera encompassing 222 species”, which is a rather static way of introducing the phylogeny of Botryosphaeriaceae. I would rather say “It comprises at least 24 genera encompassing 222 species”, or “It comprises at least 24 known genera encompassing 222 known species”.

Overall, I find the manuscript clearer as it is and have only one major suggestion to make this manuscript in the standard for publication. It concerns the pathogenicity test, the statistical analysis conducted on it and its interpretation. First of all, I suggest to add in the material and method (Pathogenicity test section) that lesion size was used as a proxy to measure the aggressiveness of the isolate. As you mentioned to one of the other reviewer comment, aggressiveness is a complex trait and proxies are often used, and are always good to define in the text. Then, the "Statistical analyzes" section from the Materials and Methods has still a margin of improvement. In your case, the ANOVA was used to test the genotype effect on the lesion size variation, with significance accepted for P<0.05. The test used to test the normality of the data could be also presented. Finally, what you can say so far is that some isolates belonging to some species was more aggressive than other isolates, belonging maybe to other species. But, you didn’t test the species effect, which would appreciate the variation of the means per species and test if the means are different. Hence you cannot say that one species is more aggressive than another one (L403 in the discussion, and the whole paragraph suggest so as it is presented). If the standard deviation is too important (D. seriata for example), therefore you’ll need more genotype per species to reach statistical significance and confirm those species effects, and then used HSD to identify which species is statistically different from the others. Therefore, I suggest to adapt the discussion in consequence, or add more genotypes in the pathogenicity test if possible.

Finally, I hope this section will be more accurately presented and this manuscript will be published as the results from this work will be beneficial for the community.

Minor comments:

I’m surprised by the answer of one of my comment about the basidiocarp observed in some citrus orchard:

“Basidiocarps are the fruiting bodies of the decay-causing fungi. Their presence on the trunk means

that it is an already rotting trunk and that some ascomycetes (Botryosphaeriaceae, Diatrypaceae…)

and Basidiomycetes (Fomitiporia, Phellinus….) have already colonized the trunk.”

Is that always true? For me, some decay-causing fungi are able to provoke the rotting of the trunk by

themselves (e.g. Heterobasidion on pine trees).

L95: There may have a misunderstanding in my comment. I was more interested in the GPS location of the orchards than the GPS location of the city. If the authors don’t have access to this information, this is not a big deal. The aim of my comment was to give additional information that could feed the discussion about the distribution pattern of the Bot species. Was L. mitidjana sp. nov. isolated both year? In similar orchard (tree ages?)

L374: “L. mitidjana sp. nov. was found in the five surveyed orchards. “ suggests that L. mitidjana sp. nov. was isolated from all the surveyed orchards. I suggest “L. mitidjana sp. nov. was found on five orchards out of ten”, or “L. mitidjana sp. nov. was found in five of the surveyed orchards”. Furthermore, do you have some elements to discuss the distribution of the species? Is there any particularity on these locations? The table containing all the information about species distribution and presented in response to my comment is particularly interesting. I’m surprised this table is not presented, at least in supplemental file.

L397: “our country” is maybe to personal

Reviewer #3: Dear Authors

I found this piece of work really interesting, well done, with a very good language, robust methodology and, much important, the article has been strongly improved after the first revisions.

A very few small notes are included in the attached pdf.

Furthermore, my only big concern, as mentioned by another reviewer, is about the description of the new Lasiodiplodia species. But I have read what you authors already have replied to the first reviewer and I can understand your point of view.

So, I accept your reasons and your determination to consider this as a novel species.

Probably, just to be clearer in the manuscript, it could be useful to sequence the other fundamental locus for the Botryosphaeriaceae identification, tubulin.

Of course it would be crazy to re-open the phylogenetic analysis, but at least, right because you consider it a new species, having the TUB sequences deposited in Genbank would be nice for future studies. And maybe, you can also mention and discuss something about nucleotide differences in TUB between your new species and L. citricola.

But I confirm, the analysis and phylogenetic tree are good and I consider them great to achieve the aim of this study.

7. PLOS authors have the option to publish the peer review history of their article (what does this mean?). If published, this will include your full peer review and any attached files.

Reviewer #1: Yes: Benoit LAURENT

Reviewer #3: No

---

## [Author Response · Author response to Decision Letter 1]

19 Mar 2020

Answers to the academic editor

I recommend starting out the introduction with a wider scope. Citrus cultivation is not only important in Algeria but in many countries and the fruits are being exported and eaten in even more countries, therefore I would start out with some sentences about the importance of Citrus in general. This could make the publication more attractive to a wider readership.

R: Revised as recommended

The authors mention at the end of the introduction that Botryosphaeriaceae have not been studied in detail in Algeria. Have other pathogen families or groups on Citrus already been studied in more detail? I think this should be introduced here in detail! This could also explain why the authors only focus on Botryosphaeriaceae and not check which other pathogens are causing the symptoms in their samples. Please explain that in more detail!

R: In this part of the introduction, we mentioned that there has not been any thorough study about citrus in Algeria. Until today, trunk diseases caused by Botryosphaeriaceae species has been widely reported in Algeria and has been associated with many important crops such as grapevine, date palm and some other fruit and forest trees. This fact led us to focus only on Botryosphaeriaceae in this study. 

Regarding citrus dieback, the only previous study done so far is the one published by Linaldeddu et al. (2015) where we described Lasiodiplodia mediterranea based on few samples. Thus, in the present study, we decided to collect more samples from several orchads in order to sudy the diversity of the Botryosphaeriaceae and their association with citrus diaback. Indeed, this study represents the most comprehensive study on the presence, diversity and pathogenicity of Botryosphaeriaceae species associated with declining citrus trees in Algeria.

The objectives at the end of the introduction should be described in more detail! The authors do not even mention that they also aimed at testing the aggressiveness of the the isolates!

R: Revised as recommended

Table 1, I agree with Benoit Laurent that it would be relevant (if possible) to add the GPS coordinates of each orchard in each region as it would give an idea of how far apart the orchards are. If this information cannot be given, I would rather remove the GPS coordinates of the cities.

R: Revised as recommended (Unfortunately, we were not able to get the exact GPS coordinates of each orchard. Thus, as advised, we removed the GPS coordinates from the table). 

l. 145 Mention which was the best fitting DNA evolution model.

R: Revised as recommended

Please improve the description of the statistical analyses and test specifically for differences in aggressiveness between species as suggested by Benoit Laurent.

R: Revised as recommended

I am a little surprised about the paragraph called “Nomenclature”. I am not familiar with the description of new fungal species but I have never seen such a justification. Is this typical? Otherwise, it should be removed. I suggest to name this paragraph “Taxonomy” which should be mentioned that it has been described in detail and that the new names have been submitted to MycoBank.

R: Regarding the Nomenclature, we included it because it was one of the requirements of the journal, for the description of new species. I believe that adding it or removing it all depends on the choice of the editor. I checked two papers, one of them has it and the other does not. 

In the results the authors mention that they obtained 47 fungal colonies belonging to Botryosphaeriaceae. Did they also observe other fungi in their isolates? What happened to the other samples?

R: In this study, Botryosphaeriaceae species accounted for more than 50% of the isolates, making this by far the most common taxonomic group associated with branch cankers in declining citrus trees. As mentioned previously in the introduction, we wanted to focus our study only on Botryosphaeriaceae due to their important impact worldwide. Regarding the remaining isolates, in addition to the wood samples which did not develop any fungus, we obtained the following genera: Alternaria, Pestalotiopsis and some sterile fungi, which we could not identify. These samples were not considered in the study due to their low frequency as well as to our focus on Botryosphaeriaceae.

l. 41 clementines

R: Revised as recommended

l. 62-64 This sentence is unclear. Do the authors refer to a single species of the Botryosphaeriaceae family or several that are all important pathogens? Please clarify!

R: Towards the end of the same paragraph, we have mentioned all the Botryosphaeriaceae species belonging to several genera that impact the citrus, as described in different studies. 

l. 77 …citrus trees in Algeria…

R: Revised as recommended

l. 350 …similar situations have been described…

R: Revised as recommended

l. 351 …according to some authors… is not a good formulation!

R: Revised as recommended (we replaced it by, ‘According to previous studies’)

Answers to the reviewer 1

General comments:

Reviewer #1: I would like to thank Akila Berraf-Tebbal and his collaborators to the answers they brought to my questions and comments. The data on the tef1 alpha are now available, which was one major concern of my last review, and allow me to verify the monophyletic group corresponding to the introduction of a new species. The debate on this introduction is still questionable but does not question the quality and the presentation of the work. I don’t think that following a trend is really constructive and yet scientific, but I endorse their point of view about the dynamicity of the taxonomy. 

Besides, I’m rather surprised that, in response to my comment about the lack of general information on Botryosphaeriaceae, a sentence is brought from a reference and citing “It comprises 24 genera encompassing 222 species”, which is a rather static way of introducing the phylogeny of Botryosphaeriaceae. I would rather say “It comprises at least 24 genera encompassing 222 species”, or “It comprises at least 24 known genera encompassing 222 known species”.

R: Revised as recommended

Overall, I find the manuscript clearer as it is and have only one major suggestion to make this manuscript in the standard for publication. It concerns the pathogenicity test, the statistical analysis conducted on it and its interpretation. First of all, I suggest to add in the material and method (Pathogenicity test section) that lesion size was used as a proxy to measure the aggressiveness of the isolate. As you mentioned to one of the other reviewer comment, aggressiveness is a complex trait and proxies are often used, and are always good to define in the text. 

R: Revised as recommended (we mentioned that internal lesion length was used as a proxy to measure the aggressiveness of the isolates)

Then, the "Statistical analyzes" section from the Materials and Methods has still a margin of improvement. In your case, the ANOVA was used to test the genotype effect on the lesion size variation, with significance accepted for P<0.05. The test used to test the normality of the data could be also presented. Finally, what you can say so far is that some isolates belonging to some species was more aggressive than other isolates, belonging maybe to other species. But, you didn’t test the species effect, which would appreciate the variation of the means per species and test if the means are different. Hence you cannot say that one species is more aggressive than another one (L403 in the discussion, and the whole paragraph suggest so as it is presented). If the standard deviation is too important (D. seriata for example), therefore you’ll need more genotype per species to reach statistical significance and confirm those species effects, and then used HSD to identify which species is statistically different from the others. Therefore, I suggest to adapt the discussion in consequence, or add more genotypes in the pathogenicity test if possible.

Finally, I hope this section will be more accurately presented and this manuscript will be published as the results from this work will be beneficial for the community.

R: Revised as recommended (The Shapiro-Wilk test conducted for normality checking was presented. As for the species effect, we agree with the reviewer that we need to conduct further tests on other isolates for each species in order to get clear conclusion about differences in aggressiveness between species. Unfortunately, due to some circumstances, it would be almost impossible to perform these tests. Thus, we made sure to adapt the discussion accordingly and take into consideration the reviewer’s comments.)

Minor comments:

I’m surprised by the answer of one of my comment about the basidiocarp observed in some citrus orchard:“Basidiocarps are the fruiting bodies of the decay-causing fungi. Their presence on the trunk means that it is an already rotting trunk and that some ascomycetes (Botryosphaeriaceae, Diatrypaceae…) and Basidiomycetes (Fomitiporia, Phellinus….) have already colonized the trunk.” Is that always true? For me, some decay-causing fungi are able to provoke the rotting of the trunk by themselves (e.g. Heterobasidion on pine trees).

R: I agree with the reviewer. What I meant, back in the first review, is that in our case, all the basidiocarps were observed in already rotting trunk. Indeed, it is true that it is not always the case. As the reviewer mentioned, some decay-causing fungi are able to provoke the rotting of the trunk by themselves. 

L95: There may have a misunderstanding in my comment. I was more interested in the GPS location of the orchards than the GPS location of the city. If the authors don’t have access to this information, this is not a big deal. The aim of my comment was to give additional information that could feed the discussion about the distribution pattern of the Bot species. Was L. mitidjana sp. nov. isolated both year? In similar orchard (tree ages?)

R: Regarding the GPS coordinates, unfortunately, we were not able to obtain the exact coordinates of the orchards. As advised by the editor, we will remove the city coordinates since they are not relevant. Regarding the isolates, different samples from different orchards have been collected during the time period from 2013 to 2015. Thus, replying to question of the reviewer, we did not collect samples from the same orchard each year. All the prospected orchards had approximately the same age (between 25 and 30 years old).

L374: “L. mitidjana sp. nov. was found in the five surveyed orchards. “ suggests that L. mitidjana sp. nov. was isolated from all the surveyed orchards. I suggest “L. mitidjana sp. nov. was found on five orchards out of ten”, or “L. mitidjana sp. nov. was found in five of the surveyed orchards”. 

R: Revised as recommended

Furthermore, do you have some elements to discuss the distribution of the species? Is there any particularity on these locations? The table containing all the information about species distribution and presented in response to my comment is particularly interesting. I’m surprised this table is not presented, at least in supplemental file.

R: Revised as recommended (the table is in the supplementary files)

L397: “our country” is maybe to personal

R: Revised as recommended (we replaced it by Algeria)

Answers to the reviewer 3

Reviewer #3: Dear Authors

I found this piece of work really interesting, well done, with a very good language, robust methodology and, much important, the article has been strongly improved after the first revisions.

A very few small notes are included in the attached pdf.

R: Revised as recommended

Furthermore, my only big concern, as mentioned by another reviewer, is about the description of the new Lasiodiplodia species. But I have read what you authors already have replied to the first reviewer and I can understand your point of view.

So, I accept your reasons and your determination to consider this as a novel species.

Probably, just to be clearer in the manuscript, it could be useful to sequence the other fundamental locus for the Botryosphaeriaceae identification, tubulin.

Of course it would be crazy to re-open the phylogenetic analysis, but at least, right because you consider it a new species, having the TUB sequences deposited in Genbank would be nice for future studies. And maybe, you can also mention and discuss something about nucleotide differences in TUB between your new species and L. citricola.

But I confirm, the analysis and phylogenetic tree are good and I consider them great to achieve the aim of this study.

R: Thank you for your comments and for understanding. We agree with your point. We could sequence the TUB and deposit it in GenBank and have them available for future studies. To be honest, in the meantime, it is hard to sequence the TUB due to the present circumstances and all what is happening around the word. We agree that this may be debatable. In fact, we have discussed this previously with the second reviewer. However, our point is that our work is clearly aligned with the current trend for introduction of novel Lasiodiplodia species. In fact, in all studies, TEF1 is considered as the most informative locus used to ID species for the Botryosphaeriaceae. We understand that the taxonomy is dynamic and hence frequently changing. Yet, for the moment we would like to introduce the new species. The fact is that eventually it will be described as a new species, if not by our group, then by someone else. This is why, if possible, we would appreciate if the acceptance for publication of this manuscript is not delayed because of that.

As an example of a case which is similar to ours, here are the nucleotide differences for the following mentioned species: L. chinensis vs L. lignicola vs L. pseudotheobromae. For all 3 species ITS is 100 % identical. We aligned the TUB sequences of these species and obtained 100% of similarity between them. As for Tef1: L. chinensis vs L. lignicola: 1 nucleotide difference L. chinensis vs L. pseudotheobromae: 3 nucleotides differences

The authors cite the articles number 8,9,10,11. I don’t think the paper number 11 is related with Diaporthe in Europe. It should be removed. On the contrary, recent studies on DIaporthe which should be cited are:

-Guarnaccia, V., & Crous, P. W. (2018). Species of Diaporthe on Camellia and Citrus in the Azores Islands. Phytopathologia Mediterranea

-Guarnaccia, V., & Crous, P. W. (2017). Emerging citrus diseases in Europe caused by species of Diaporthe. IMA fungus, 8(2), 317-334.

R: Revised as recommended (we removed the reference 11 and added the suggested ones)

Personally, I agree with this atypical inoculatin way, I mean for the cut twigs. However, it should be better if the authors support this method with other article(s) where the same method has been used on Citrus spp.

R: Revised as recommended

Please start the sentence with the entire name Lasiodiplodia. Do not abbreviate the first word.

R: In this particular line, we did not include the full name since it has already been cited using the entire name when it has been first mentioned in the manuscript.

---

## [Editor Report · Decision Letter 2]

16 Apr 2020

Lasiodiplodia mitidjana sp. nov. and other Botryosphaeriaceae species causing branch canker and dieback of Citrus sinensis in Algeria

PONE-D-19-29810R2

Dear Dr. BERRAF-TEBBAL,

We are pleased to inform you that your manuscript has been judged scientifically suitable for publication and will be formally accepted for publication once it complies with all outstanding technical requirements.

With kind regards,

Katharina B Budde, Ph.D.

Academic Editor

PLOS ONE

Additional Editor Comments:

The manuscript “*Lasiodiplodia mitidjana *sp. nov. and other Botryosphaeriaceae species causing branch canker and dieback of *Citrus sinensis* in Algeria” has received two rounds of review. The authors have satisfactorily taken into account the comments raised by the three independent reviewers and myself and I would like to congratulate the authors for the revised manuscript. Concerning the paragraph on the “Nomenclature”, I agree with the authors to include it, as it conforms to the guidelines of PLOS ONE. The methodology is clear and concise and the manuscript has improved considerably during the revision process, therefore I recommend publication. However, the following very minor comments should still be taken into account in the final version of the manuscript.

l. 152-153: Not clear! Maybe: A discrete Gamma distribution with five categories was used to model evolutionary rate differences among sites.

l. 194-195 Please insert the names of the digital repositories. All PLOS articles are deposited in PubMed Central and LOCKSS. If your institute, or those of your co-authors, has its own repository, we recommend that you also deposit the published online article there and include the name in your article.

l. 420 …contradict a previous study…

l.421 Did the authors in reference 62 compare the aggressiveness of the same species as in the present study? Please mention this here.

I did not notice before but typically, family names, such as Botryosphaeriaceae are not written in itallics, please correct.

---

## [Editor Report · Acceptance letter]

24 Apr 2020

PONE-D-19-29810R2 

*Lasiodiplodia mitidjana* sp. nov. and other Botryosphaeriaceae species causing branch canker and dieback of *Citrus sinensis* in Algeria 

Dear Dr. Berraf-Tebbal:

I am pleased to inform you that your manuscript has been deemed suitable for publication in PLOS ONE. Congratulations! Your manuscript is now with our production department. 

With kind regards,

on behalf of

Dr. Katharina B Budde 

Academic Editor

PLOS ONE